# SSNet: Skip and Split MLP Network for Long-Term Series Forecasting

## Abstract

Time series forecasting is critical across various domains, including energy, transportation, weather prediction, and healthcare. Although recent advances using CNNs, RNNs, and Transformer-based models have shown promise, these approaches often suffer from architectural complexity and low computational efficiency. MLP-based networks offer better computational efficiency, and some frequency-domain MLP models have demonstrated the ability to handle periodic time series data. However, standard MLP-based methods still struggle to directly model periodic and temporal dependencies in the time domain, which are essential for accurate time series forecasting. To address these challenges, we propose the Skip and Split MLP Network (**SSNet**), featuring innovative Skip-MLP and Split-MLP components that enable MLP models to directly capture periodicity and temporal dependencies in the time domain. SSNet requires fewer parameters than traditional MLP-based architectures, improving computational efficiency. Empirical results on multiple real-world long-term forecasting datasets demonstrate that SSNet significantly outperforms state-of-the-art models, delivering better performance with fewer parameters. Notably, even a single Skip-MLP unit matches the performance of high-performing models like PatchTST.

## 1 Introduction

Time series forecasting has been extensively applied in various practical scenarios, including economics (Clements et al., 2004; Ghysels & Marcellino, 2018), energy (Deb et al., 2017; Koprinska et al., 2018; Hu et al., 2022), transportation (Cirstea et al., 2022; Cai et al., 2020), weather (Hewage et al., 2020; Taylor et al., 2009), and retail (Fildes et al., 2022). In particular, Long-Term Time Series forecasting presents significant real-world demands, where accurate modeling of periodic and seasonal patterns is crucial for improving performance. PatchTST (Nie et al., 2022) partitions time series into multiple small patches to enhance the information of local sequence features and reduce the computational cost associated with excessively long sequences. TimesNet (Wu et al., 2022) uses the Fast Fourier Transform to perform multi-scale decomposition of time series data and processes the 2D feature maps with CNN networks to capture rich periodic information. TimeMixer (Wang et al., 2024) leverages down-sampling techniques to hierarchically process sequences and aggregates seasonal and trend information through Seasonal Mixing and Trend Mixing. These methods utilize different approaches to capture periodic information and model both local and global information. DLinear (Zeng et al., 2023), based on trend and seasonality decomposition, employs a single linear layer to demonstrate the MLP-based model's excellent capability in time series forecasting. However, due to its symmetric point-mapping structure, MLPs struggle to model periodic and seasonal patterns directly and are limited in distinguishing the temporal dependencies within sequences.

To address these limitations, we propose two novel network architectures, **Skip-MLP** and **Split-MLP**, which extend traditional MLPs. These architectures consist of fundamental **Skip Connected Layers** and **Split Connected Layers**, as illustrated in Figure 1. Skip-MLP skips over sequences of a specific length (referred to as skip-size, generally related to the sequence's periodicity) to capture periodic information. The Split-MLP layer segments sequences into sub-sequences of specific lengths, establishing separate connections between these sub-sequences to focus on extracting local features. When used in conjunction with Skip-MLP layers, it can further capture global sequence information. Through matrix derivation, we demonstrate that the sparse Skip Connection Layer and Split Connected Layer can be represented as combinations of multiple small, dense Fully Connected

Layers. Experimental results further reveal that sharing parameters among these small matrices enhances prediction performance and computational efficiency.

To leverage the strengths of both Skip-MLP and Split-MLP layers, we designed the **SSNet** network architecture, which includes three components: (1) an **Auto-correlation Block** that adaptively extracts significant periodic patterns and their intensities, (2) a parallel multi-scale **Skip-MLPs** network for capturing and integrating diverse periodic information, and (3) a serial **SS-MLP Block** with residual connections that progressively reduces prediction bias. In our experiments, SSNet consistently achieved state-of-the-art performance across various real-world forecasting scenarios while maintaining linear time complexity. Our contributions are summarized as follows:

- We introduce two novel MLP networks, Skip-MLP and Split-MLP, which address the limitations of traditional MLPs in capturing periodic information and can handle global and local data information more effectively with fewer parameters.
- We propose the SSNet model architecture, which adaptively captures and integrates critical periodic patterns, effectively differentiating between local and global information, making it well-suited for long-term multivariate time series forecasting.
- Extensive experimental results demonstrate that SSNet achieves state-of-the-art performance in long-term time series forecasting tasks, with fewer parameters and higher computational efficiency.

## 2 RELATED WORK

**Research on MLPs**    Multi-Layer Perceptrons (MLPs) are fundamental structures in deep learning and have been extensively studied. The MLP-Mixer(Tolstikhin et al., 2021) achieves competitive performance in image classification by employing Token-mixing MLPs and Channel-mixing MLPs to extract features from positional tokens and channels, respectively, thereby demonstrating the efficient visual representation capabilities of MLPs. The gMLP(Liu et al., 2021) introduces gating mechanisms to simulate the self-attention mechanisms found in Transformers, achieving performance that is competitive with or even exceeds that of Transformers in both language and vision tasks. ResMLP(Touvron et al., 2022) simplifies networks based on Vision Transformers (ViT)(Dosovitskiy, 2020) by incorporating MLPs and residual connections, leading to high-performance image classification. sMLPNet(Tang et al., 2022) utilizes 1D MLPs applied axially, with parameter sharing between rows or columns, which reduces the number of model parameters and computational complexity through sparse connections and weight sharing. Recently, the KAN(Liu et al., 2024b) model has sparked considerable discussion due to its superior interpretability and improved performance in certain prediction scenarios, positioning it as a strong contender among MLP variants. We propose enhancements to the traditional MLP architecture through Skip Connected Layers and Split Connected Layers, which enhance its ability to handle sequential data and improve computational efficiency.

**Time Series Forecasting**    Time series prediction has been extensively studied, with traditional methods such as Exponential Smoothing(Gardner Jr, 1985), Holt-Winters(Chatfield, 1978) and ARIMA(Asteriou & Hall, 2011) providing robust theoretical foundations and good interpretability. However, these methods are limited by their assumptions and struggle with the complexities of real-world time series data. Recent research has increasingly focused on deep learning-based approaches for time series forecasting. RNN-based models (LSTNet(Lai et al., 2018), DeepAR(Salinas et al., 2020)) capture sequential dependencies but encounter challenges such as gradient vanishing and forgetting in long sequences. CNN-based models (MICN(Wang et al., 2023), SCINet(Liu et al., 2022a), TimesNet(Wu et al., 2022)) leverage CNN characteristics to capture 2D features but are often limited by receptive field sizes, impacting global feature extraction. Transformer-based models (PatchTST(Nie et al., 2022), FEDFormer(Zhou et al., 2022b), Stationary(Liu et al., 2022b)) benefit from self-attention mechanisms to extract rich temporal features and show strong prediction performance, though they face high computational complexity with longer sequences. MLP-based models (LTST-Linear(Zeng et al., 2023), TimeMixer(Wang et al., 2024), NBEATS(Iori et al., 2024)) are generally simpler and more computationally efficient but struggle with directly modeling periodic and local features. We propose the SSNet architecture for time series data modeling, which adaptively performs multi-scale modeling, effectively integrates both local and global information, and achieves high computational efficiency and predictive performance.

# 3 SSNet Framework

**Problem Definition**  Multivariate time series prediction involves forecasting future values for a window of size $H$, given historical data from a look-back window of size $L$. A time series with $T$ time steps and $N$ variables is represented as $[X_1, X_2, \ldots, X_T] \in \mathbb{R}^{N \times T}$, where $X_t \in \mathbb{R}^N$ denotes the $N$-dimensional values at time step $t$. Given the current time step $t$, the input sequences fed into the model is $\mathbf{X_t} = [X_{t-L+1}, X_{t-L+2}, \ldots, X_t] \in \mathbb{R}^{N \times L}$, and the output will be $\mathbf{Y_t} = [X_{t+1}, X_{t+2}, \ldots, X_{t+H}] \in \mathbb{R}^{N \times H}$.

## 3.1 Skip Connected Layer and Split Connected Layer

**Connection Mechanisms**  Traditional Multilayer Perceptrons (MLPs) are typically composed of multiple Fully Connected Layers, which ensure that units in adjacent layers are entirely interconnected. This architecture enables the network to model complex functions through successive layers. However, is this the most optimal configuration? In the context of time series forecasting, we introduce two novel connection mechanisms: the **Skip Connected Layer** and the **Split Connected Layer**, as illustrated in Figure 1.

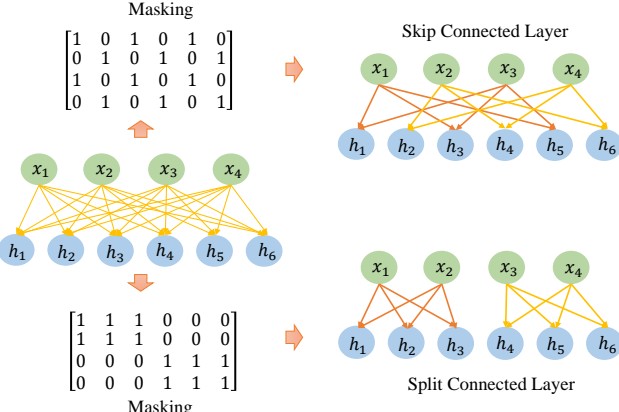

Figure 1: Examples of Skip Connected Layer and Split Connected Layer. A Skip Connected Layer or Split Connected Layer is equivalent to applying a masking matrix on a Fully Connected Layer. The figure shows the cases where skip-size and split-size are both equal to 2.

For a fully connected layer, given an input sequence $\mathbf{X_t} \in \mathbb{R}^{N \times L}$ and a hidden layer dimension $D$, the output is:

$$\mathbf{O_t} = \mathbf{X_t} W_{Fully} + B_{Fully} = [O_1, O_2, \ldots, O_D] \in \mathbb{R}^{N \times D}$$

where

$$W_{Fully} \in \mathbb{R}^{L \times D}, B_{Fully} \in \mathbb{R}^{1 \times D}$$

For a Skip Connected Layer with a skip size $S$, define $N_L = \lceil \frac{L}{S} \rceil$ and $N_D = \lceil \frac{D}{S} \rceil$ as the numbers of input and output groups respectively, $\lceil \cdot \rceil$ denotes the ceiling function. To align input and output dimensions with the skip size, zero padding is applied where necessary. The output can be expressed as:

$$\mathbf{O_t} = \mathbf{X_t} W_{Skip} + B_{Skip} = [O_1, O_2, \ldots, O_{N_D \times S}] \in \mathbb{R}^{N \times N_D \cdot S}$$

where

$$W_{Skip} = W_{Fully} \odot W_{Mask} \in \mathbb{R}^{N_L \cdot S \times N_D \cdot S}, \quad B_{Skip} \in \mathbb{R}^{1 \times N_D \cdot S}$$

$$W_{Mask_{ij}} = \begin{cases} 1 & \text{if } |i - j| \bmod S = 0 \\ 0 & \text{otherwise} \end{cases}$$

Here, $W_{Mask}$ is a sparse matrix, transforming the Fully Connected Layer into a Skip Connected Layer, the matrix value is 1 if the absolute difference between indices is a multiple of $S$, and 0 otherwise. With appropriate matrix transformations and data compression, the Skip Connected Layer

can be represented as a concatenation of several small dense matrices (proof provided in the Appendix A). Here, multiple $O_{t_i}$ together form the output $\mathbf{O_t}$:

$$O_{t_i} = X_{t_i} W_{Fully_i} + B_i = \left[ O_i, O_{i+S}, \ldots, O_{i+(N_D-1)S} \right] \in \mathbb{R}^{N \times N_D}$$

$$X_{t_i} = \left[ X_i, X_{i+S}, \ldots, X_{i+(N_L-1)S} \right] \in \mathbb{R}^{N \times N_L}$$

$$W_{Fully_i} \in \mathbb{R}^{N_L \times N_D}, \quad B_i \in \mathbb{R}^{1 \times N_D}, \quad i \in \{1, 2, \ldots, S\}$$

Similarly, for a Split Connected Layer with a split size $S$, define $N_L = N_D = \left\lceil \frac{L}{S} \right\rceil$ and $S_D = \left\lceil \frac{D}{N_D} \right\rceil$ as the length of each output group. After transformation, a simplified form can be obtained:

$$\mathbf{O_t} = \left[ O_{t_1}, O_{t_2}, \cdots, O_{t_{N_D}} \right] \in \mathbb{R}^{N \times N_D \cdot S_D}$$

$$O_{t_i} = X_{t_i} W_{Fully_i} + B_i = \left[ O_i, O_{i+1}, \ldots, O_{i+S_D-1} \right] \in \mathbb{R}^{N \times S_D}$$

$$X_{t_i} = \left[ X_i, X_{i+1}, \ldots, X_{i+S-1} \right] \in \mathbb{R}^{N \times S}$$

$$W_{Fully_i} \in \mathbb{R}^{S \times S_D}, \quad B_i \in \mathbb{R}^{1 \times S_D}, \quad i \in \{1, 2, \ldots, N_D\}$$

Thus, in practical implementations, these connection mechanisms can be realized by concatenating multiple small dense networks.

### 3.2 CONSTRUCTION OF SKIP-MLP AND SPLIT-MLP

Building on the multilayer Skip Connected Layer and Split Connected Layer, we develop the **Skip-MLP** and **Split-MLP** architectures, as depicted in Figure 2. As demonstrated in Section 3.1, both connection mechanisms can be modeled as multiple smaller fully connected layers with equivalent dimensions. Our experiments indicate that sharing weights among these smaller layers enhances network performance.

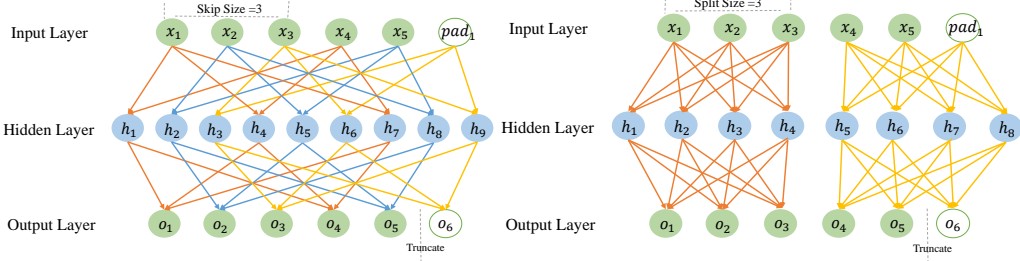

Figure 2: The left side shows an example of Skip-MLP with a skip-size of 3, while the right side depicts an example of Split-MLP with a split-size of 3. Padding is applied to handle inputs that are not divisible by the skip-size or split-size, and excess outputs are truncated. Connections with the same color represent individual small MLPs, which together form the entire network.

We integrate Skip-MLP and Split-MLP networks to form the core processing unit for temporal features, termed the **SS-MLP Block**. This block comprises two Skip-MLP layers and two Split-MLP layers.

The initial Skip-MLP layer extracts periodic sequence features and reconstructs them, preserving the original temporal order while incorporating new skipped periodic information. Subsequently, the Split-MLP layer aggregates features, capturing local information and integrating the periodic features

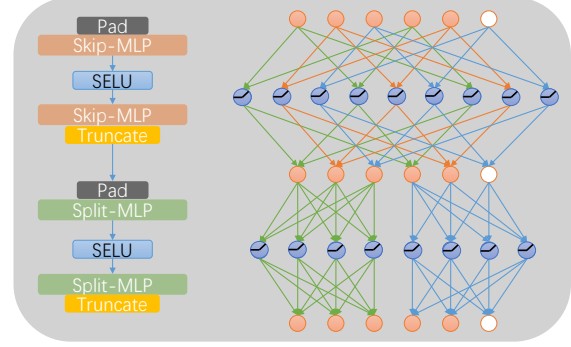

Figure 3: The overall structure of SS-MLP Block.

previously acquired. This ensures that each output unit indirectly incorporates information from all input units.

The first Skip-MLP and Split-MLP layers utilize the SELU activation function (Klambauer et al., 2017) to introduce non-linearity and enhance the model's representational capacity. We observed that SELU significantly outperforms other activation functions such as ReLU (Nair & Hinton, 2010) and GELU (Hendrycks & Gimpel, 2016) in this context.

### 3.2.1 SSNET

The SSNet architecture, depicted in Figure 5, comprises three primary components: the Auto-correlation Block, Skip-MLPs at the input layer, and SS-MLPs in the intermediate layers.

**Auto-correlation Block** The Auto-correlation Block is designed to extract periodic hyperparameters from the sequence before it enters the network. These hyperparameters are then used to configure the network structure. As illustrated in Figure 4, the block first decomposes multivariate time series data into univariate sequences, normalizes them, and computes the autocorrelation function:

$$R_{ff}(\tau) = \int_{-\infty}^{\infty} f(t+\tau)\overline{f(t)}\, dt = \int_{-\infty}^{\infty} f(t)\overline{f(t-\tau)}\, dt$$

where $f(t)$ represents the series data, $\overline{f(t)}$ denotes the complex conjugate of $f(t)$ and $\tau$ represents the lag order. $R_{ff}(\tau)$ indicates the correlation strength, with a larger value signifying stronger correlation. Strong correlation means that the feature of the sequence $f$ at time $t$ will reappear at time $t + \tau$. The discrete form is given by

$$R_{ff}(\tau) = \sum_{t\in\mathbb{Z}} f(t)\overline{f(t-\tau)}$$

For a univariate sequence $X_{it} = f_i(t) \in \mathbb{R}^{1\times L}$, we compute the autocorrelation function:

$$ACF(X_{it}) = [R_{f_i f_i}(-L+1), \ldots, R_{f_i f_i}(0), \ldots, R_{f_i f_i}(L-1)] \in \mathbb{R}^{1\times(2L-1)}$$

Local peaks from the segment $[R_{f_i f_i}(0), R_{f_i f_i}(1), \ldots, R_{f_i f_i}(L-1)]$ are identified as periodic candidates. Aggregated periodic correlations from different variables yield the top $K$ periods and their corresponding correlation strengths.

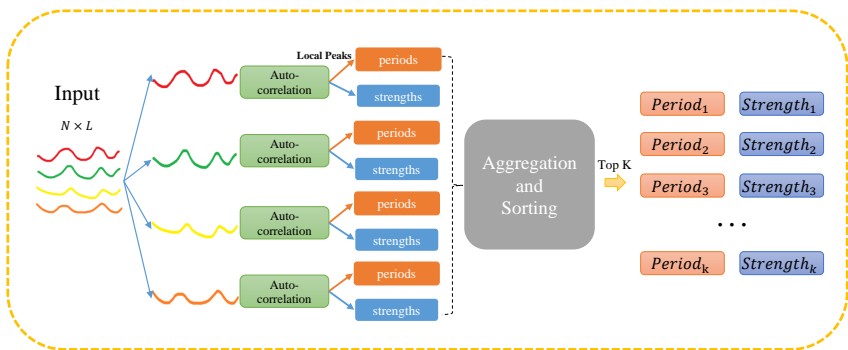

Figure 4: Auto-correlation Block.

Autocorrelation-based periods offer practical advantages over Fast Fourier Transform (FFT) (Chatfield & Xing, 2019; Zhou et al., 2022b). The top periods typically align with meaningful periodicities (e.g., hour, half-day, day, week), whereas FFT often struggles to filter out interference, particularly in low-frequency components.

**Skip-MLPs** Skip-MLPs consist of parallel networks with skip sizes determined by the top $K$ periods identified by the Auto-correlation Block. The input sequence is processed through $K$ Skip-MLPs, each with a different skip size, to extract periodic information. The results are combined based on their corresponding correlation strengths:

$$L_0 = \sum_{k=1}^{K} SkipMLP_{Period_k}(\mathbf{X_t}) \cdot Strength_k$$

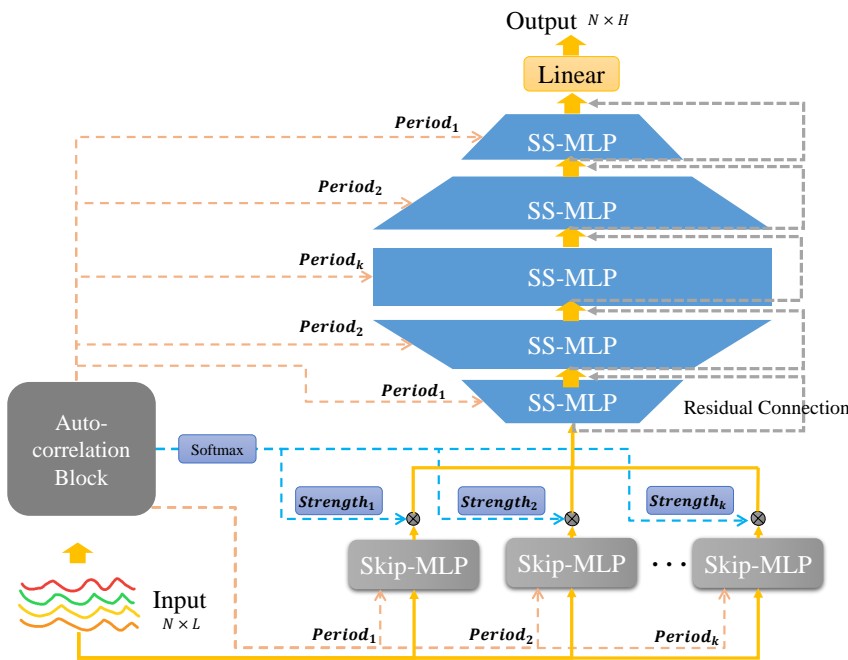

Figure 5: Overall structure of SSNet.

Here, $K$ represents the number of selected top periods, while $Period_k$ and $Strength_k$ denote the $k$-th period value and its associated correlation strength, respectively.

**SS-MLPs** SS-MLPs are composed of $2K - 1$ SS-MLP Blocks, with residual connections applied between consecutive layers. The output of the $n$-th layer is the sum of the SS-MLP output at that layer and the output of the $n - 1$-th layer. This design ensures that features extracted across multiple scales are preserved during deep propagation. Empirical results indicate that SS-MLPs achieve better performance when the input and output layers utilize smaller skip sizes. To fully capture periodic information, we adopt a design where the skip sizes are narrower at both ends and wider in the middle. This approach outperforms purely increasing or decreasing skip-size strategies, introduces no additional hyperparameters, and ensures a well-balanced, elegant network architecture.

The first $K$ layers, where the periods increase, are defined as:

$$L_n = SSMLP_{Period_n}(L_{n-1}) + L_{n-1}, \quad 1 \leq n \leq K$$

The remaining $K - 1$ layers, where the periods decrease, are defined as:

$$L_n = SSMLP_{Period_{2K-n}}(L_{n-1}) + L_{n-1}, \quad K + 1 \leq n \leq 2K - 1$$

The final output layer is implemented as a fully connected linear projection. Unlike Skip-Connected and Split-Connected Layers, the fully connected layer excels at feature aggregation, effectively consolidating the diverse periodic, global, and local features extracted by the preceding layers. This comprehensive integration enhances the prediction performance, enabling the model to achieve optimal results. The output layer maps the sequence to the target prediction length $H$:

$$\mathbf{O_t} = Linear(L_{2K-1})$$

## 4 EXPERIMENTS

**Datasets** To evaluate the performance of the SSNet model, experiments were conducted on several widely used real-world long-sequence datasets (Zhou et al., 2021). These include the ETT dataset (comprising four subsets: ETTh1, ETTh2, ETTm1, ETTm2), as well as Weather, Electricity, and Traffic datasets. A more detailed description of each dataset is provided in the Appendix B. The basic characteristics for each dataset are summarized below:

Table 1: Details of the datasets used for evaluation.

| Datasets | ETTh1 | ETTh2 | ETTm1 | ETTm2 | Weather | Electricity | Traffic |
|---|---|---|---|---|---|---|---|
| **Frequency** | hour | hour | 15min | 15min | 10min | hour | hour |
| **Features** | 7 | 7 | 7 | 7 | 21 | 321 | 862 |
| **Timesteps** | 17,420 | 17,420 | 69,680 | 69,680 | 52,696 | 26,304 | 17,544 |

**Baselines** We evaluate SSNet against ten state-of-the-art methods, including Transformer-based models such as PatchTST (Nie et al., 2022), PDF (Dai et al., 2024), and FEDFormer (Zhou et al., 2022b); CNN-based models like TimesNet (Wu et al., 2022), ModernTCN Luo & Wang (2024), MICN (Wang et al., 2023), and FiLM (Zhou et al., 2022a); as well as MLP-based models including TimeMixer (Wang et al., 2024) and FITS Xu et al. (2023), DLinear (Zeng et al., 2023).

**Implementation Details** All experiments were conducted on a single NVIDIA A100 80GB GPU, using MSE Loss for training. The prediction output lengths were set to $H = \{96, 192, 336, 720\}$. The look-back length followed the specifications in the original papers for each model, with SSNet using a fixed look-back window size of 512. Baseline models were optimized with their respective best hyperparameters, whereas SSNet used a consistent set of parameters across different lengths for the same dataset.

## 4.1 MAIN RESULTS

Table 2: Forecasting performance for multivariate long-term time series with prediction lengths $H = \{96, 192, 336, 720\}$. The best results are highlighted in **bold red**, while the second-best are marked with underlined blue. "Avg" represents the average performance across all lengths, and "COUNT" indicates the number of times each model achieved the best result. The numbers in parentheses indicate the look-back length for each model. Some model codes are not publicly available and results are cited from the respective papers.

| | | MLP-Based | | | | | | | | Transformer-Based | | | | | | CNN-Based | | | | | | | |
|---|---|---|---|---|---|---|---|---|---|---|---|---|---|---|---|---|---|---|---|---|---|---|---|
| Models | | SSNet (Ours, 512) | | TimeMixer (Unknown) | | FITS (Unfixed) | | DLinear (336) | | PatchTST (512) | | PDF (720) | | FEDformer (96) | | FiLM (Unfixed) | | TimesNet (96) | | ModernTCN (Unfixed) | | MICN (96) | |
| Metric | | MSE | MAE | MSE | MAE | MSE | MAE | MSE | MAE | MSE | MAE | MSE | MAE | MSE | MAE | MSE | MAE | MSE | MAE | MSE | MAE | MSE | MAE |
| ETTh1 | 96 | **0.339** | **0.374** | 0.361 | 0.390 | 0.371 | 0.394 | 0.375 | 0.399 | 0.370 | 0.400 | 0.356 | 0.391 | 0.376 | 0.419 | 0.422 | 0.432 | 0.384 | 0.402 | 0.369 | 0.394 | 0.421 | 0.431 |
| | 192 | **0.359** | **0.386** | 0.409 | 0.414 | 0.404 | 0.413 | 0.405 | 0.416 | 0.413 | 0.429 | 0.390 | 0.413 | 0.420 | 0.448 | 0.462 | 0.458 | 0.436 | 0.429 | 0.406 | 0.414 | 0.474 | 0.487 |
| | 336 | **0.370** | **0.399** | 0.430 | 0.429 | 0.425 | 0.425 | 0.439 | 0.443 | 0.422 | 0.440 | 0.402 | 0.421 | 0.459 | 0.465 | 0.501 | 0.483 | 0.491 | 0.469 | 0.392 | 0.412 | 0.569 | 0.551 |
| | 720 | 0.425 | 0.447 | 0.445 | 0.460 | **0.420** | **0.442** | 0.472 | 0.490 | 0.447 | 0.468 | 0.462 | 0.477 | 0.506 | 0.507 | 0.544 | 0.526 | 0.521 | 0.500 | 0.450 | 0.461 | 0.770 | 0.672 |
| | Avg | **0.373** | **0.402** | 0.411 | 0.423 | 0.405 | 0.419 | 0.423 | 0.437 | 0.413 | 0.434 | 0.402 | 0.426 | 0.440 | 0.460 | 0.482 | 0.475 | 0.458 | 0.450 | 0.404 | 0.421 | 0.559 | 0.535 |
| ETTh2 | 96 | **0.216** | **0.298** | 0.271 | 0.330 | 0.272 | 0.337 | 0.289 | 0.353 | 0.274 | 0.337 | 0.270 | 0.332 | 0.358 | 0.397 | 0.323 | 0.370 | 0.340 | 0.374 | 0.264 | 0.333 | 0.299 | 0.364 |
| | 192 | **0.261** | **0.332** | 0.317 | 0.402 | 0.331 | 0.375 | 0.383 | 0.418 | 0.341 | 0.382 | 0.334 | 0.375 | 0.429 | 0.439 | 0.391 | 0.415 | 0.402 | 0.414 | 0.318 | 0.373 | 0.441 | 0.454 |
| | 336 | **0.300** | **0.364** | 0.332 | 0.396 | 0.339 | 0.388 | 0.448 | 0.465 | 0.329 | 0.384 | 0.324 | 0.379 | 0.496 | 0.487 | 0.415 | 0.440 | 0.452 | 0.452 | 0.314 | 0.376 | 0.654 | 0.567 |
| | 720 | 0.372 | 0.419 | 0.342 | 0.408 | 0.372 | 0.420 | 0.605 | 0.551 | 0.379 | 0.422 | 0.378 | 0.422 | 0.463 | 0.474 | 0.441 | 0.459 | 0.462 | 0.468 | 0.394 | 0.432 | 0.956 | 0.716 |
| | Avg | **0.287** | **0.353** | 0.316 | 0.384 | 0.329 | 0.380 | 0.431 | 0.447 | 0.331 | 0.381 | 0.327 | 0.377 | 0.437 | 0.449 | 0.393 | 0.421 | 0.414 | 0.427 | 0.322 | 0.379 | 0.588 | 0.525 |
| ETTm1 | 96 | **0.277** | 0.340 | 0.291 | 0.340 | 0.304 | 0.345 | 0.299 | 0.343 | 0.293 | 0.346 | **0.277** | **0.337** | 0.379 | 0.419 | 0.302 | 0.345 | 0.338 | 0.375 | 0.297 | 0.348 | 0.316 | 0.362 |
| | 192 | **0.312** | **0.361** | 0.327 | 0.365 | 0.337 | 0.365 | 0.335 | 0.365 | 0.333 | 0.370 | 0.316 | 0.364 | 0.426 | 0.441 | 0.338 | 0.368 | 0.374 | 0.387 | 0.334 | 0.370 | 0.363 | 0.390 |
| | 336 | **0.345** | **0.380** | 0.360 | 0.381 | 0.366 | 0.385 | 0.369 | 0.386 | 0.369 | 0.392 | 0.346 | 0.381 | 0.445 | 0.459 | 0.373 | 0.388 | 0.410 | 0.411 | 0.370 | 0.392 | 0.408 | 0.426 |
| | 720 | 0.404 | **0.409** | 0.415 | 0.417 | 0.415 | 0.411 | 0.425 | 0.421 | 0.416 | 0.420 | **0.402** | 0.409 | 0.543 | 0.490 | 0.420 | 0.420 | 0.478 | 0.450 | 0.413 | 0.416 | 0.481 | 0.476 |
| | Avg | **0.335** | **0.373** | 0.348 | 0.376 | 0.355 | 0.377 | 0.357 | 0.379 | 0.353 | 0.382 | **0.335** | **0.373** | 0.448 | 0.452 | 0.358 | 0.380 | 0.400 | 0.406 | 0.353 | 0.382 | 0.392 | 0.414 |
| ETTm2 | 96 | **0.139** | **0.233** | 0.164 | 0.254 | 0.163 | 0.253 | 0.167 | 0.260 | 0.166 | 0.256 | 0.163 | 0.251 | 0.203 | 0.287 | 0.165 | 0.256 | 0.187 | 0.267 | 0.170 | 0.258 | 0.179 | 0.275 |
| | 192 | **0.180** | **0.263** | 0.223 | 0.295 | 0.217 | 0.291 | 0.224 | 0.303 | 0.223 | 0.296 | 0.217 | 0.292 | 0.269 | 0.328 | 0.222 | 0.296 | 0.249 | 0.309 | 0.227 | 0.299 | 0.307 | 0.376 |
| | 336 | **0.219** | **0.290** | 0.279 | 0.330 | 0.269 | 0.326 | 0.281 | 0.342 | 0.274 | 0.329 | 0.266 | 0.325 | 0.325 | 0.366 | 0.277 | 0.333 | 0.321 | 0.351 | 0.275 | 0.329 | 0.325 | 0.388 |
| | 720 | **0.279** | **0.332** | 0.359 | 0.383 | 0.347 | 0.377 | 0.397 | 0.421 | 0.362 | 0.385 | 0.345 | 0.375 | 0.421 | 0.415 | 0.371 | 0.389 | 0.408 | 0.403 | 0.351 | 0.381 | 0.502 | 0.490 |
| | Avg | **0.204** | **0.280** | 0.256 | 0.316 | 0.249 | 0.312 | 0.267 | 0.332 | 0.256 | 0.317 | 0.247 | 0.311 | 0.305 | 0.349 | 0.259 | 0.319 | 0.291 | 0.333 | 0.256 | 0.317 | 0.328 | 0.382 |
| Weather | 96 | **0.143** | 0.196 | 0.147 | 0.197 | **0.143** | **0.193** | 0.176 | 0.237 | 0.149 | 0.198 | 0.143 | 0.193 | 0.217 | 0.296 | 0.199 | 0.262 | 0.172 | 0.220 | 0.151 | 0.205 | 0.161 | 0.229 |
| | 192 | 0.187 | 0.240 | 0.189 | 0.239 | **0.186** | **0.236** | 0.220 | 0.282 | 0.194 | 0.241 | 0.188 | 0.236 | 0.276 | 0.336 | 0.228 | 0.288 | 0.219 | 0.261 | 0.196 | 0.247 | 0.220 | 0.281 |
| | 336 | **0.218** | **0.272** | 0.241 | 0.280 | 0.237 | 0.278 | 0.265 | 0.319 | 0.245 | 0.282 | 0.245 | 0.279 | 0.339 | 0.380 | 0.267 | 0.323 | 0.280 | 0.306 | 0.237 | 0.283 | 0.278 | 0.331 |
| | 720 | **0.297** | **0.328** | 0.310 | 0.330 | 0.307 | 0.329 | 0.323 | 0.362 | 0.314 | 0.334 | 0.308 | 0.328 | 0.403 | 0.428 | 0.319 | 0.361 | 0.365 | 0.359 | 0.315 | 0.335 | 0.311 | 0.356 |
| | avg | **0.211** | **0.259** | 0.222 | 0.262 | 0.218 | 0.259 | 0.246 | 0.300 | 0.226 | 0.264 | 0.220 | **0.259** | 0.309 | 0.360 | 0.253 | 0.309 | 0.259 | 0.287 | 0.225 | 0.267 | 0.243 | 0.299 |
| Electricity | 96 | **0.125** | **0.219** | 0.129 | 0.224 | 0.134 | 0.231 | 0.140 | 0.237 | 0.129 | 0.222 | 0.126 | 0.220 | 0.193 | 0.308 | 0.154 | 0.267 | 0.168 | 0.272 | 0.129 | 0.226 | 0.164 | 0.269 |
| | 192 | **0.143** | **0.236** | 0.140 | 0.220 | 0.148 | 0.244 | 0.153 | 0.249 | 0.145 | 0.240 | 0.147 | 0.238 | 0.201 | 0.315 | 0.164 | 0.258 | 0.184 | 0.289 | 0.143 | 0.239 | 0.177 | 0.285 |
| | 336 | **0.158** | **0.252** | 0.161 | 0.255 | 0.163 | 0.260 | 0.169 | 0.267 | 0.163 | 0.259 | 0.159 | 0.255 | 0.214 | 0.329 | 0.188 | 0.283 | 0.198 | 0.300 | 0.161 | 0.259 | 0.193 | 0.304 |
| | 720 | 0.195 | 0.286 | 0.194 | 0.287 | 0.202 | 0.293 | 0.203 | 0.301 | 0.197 | 0.290 | 0.194 | 0.287 | 0.246 | 0.355 | 0.236 | 0.332 | 0.220 | 0.320 | **0.191** | **0.286** | 0.212 | 0.321 |
| | Avg | **0.155** | 0.248 | 0.156 | 0.247 | 0.162 | 0.257 | 0.166 | 0.264 | 0.159 | 0.253 | 0.156 | 0.250 | 0.214 | 0.327 | 0.186 | 0.285 | 0.193 | 0.295 | 0.156 | 0.253 | 0.187 | 0.295 |
| Traffic | 96 | **0.347** | **0.237** | 0.360 | 0.249 | 0.386 | 0.268 | 0.410 | 0.282 | 0.360 | 0.249 | 0.350 | 0.239 | 0.587 | 0.366 | 0.416 | 0.294 | 0.593 | 0.321 | 0.368 | 0.253 | 0.519 | 0.309 |
| | 192 | 0.370 | **0.248** | 0.375 | 0.250 | 0.393 | 0.270 | 0.423 | 0.287 | 0.379 | 0.256 | **0.363** | 0.247 | 0.604 | 0.373 | 0.408 | 0.288 | 0.617 | 0.336 | 0.379 | 0.261 | 0.537 | 0.315 |
| | 336 | **0.380** | **0.255** | 0.385 | 0.270 | 0.407 | 0.277 | 0.436 | 0.286 | 0.392 | 0.264 | 0.376 | 0.258 | 0.621 | 0.383 | 0.425 | 0.298 | 0.629 | 0.336 | 0.397 | 0.270 | 0.534 | 0.313 |
| | 720 | 0.422 | **0.278** | 0.430 | 0.281 | 0.448 | 0.299 | 0.466 | 0.315 | 0.432 | 0.286 | **0.419** | 0.279 | 0.626 | 0.382 | 0.520 | 0.353 | 0.640 | 0.350 | 0.440 | 0.296 | 0.577 | 0.325 |
| | Avg | 0.380 | **0.255** | 0.388 | 0.263 | 0.408 | 0.279 | 0.434 | 0.293 | 0.391 | 0.264 | **0.377** | 0.256 | 0.610 | 0.376 | 0.442 | 0.308 | 0.620 | 0.336 | 0.396 | 0.270 | 0.542 | 0.316 |
| COUNT | | **52** | | 5 | | 7 | | 0 | | 0 | | 16 | | 0 | | 0 | | 0 | | 2 | | 0 | |

The results of multivariate long-term time series forecasting are summarized in Table 2. Across all datasets, SSNet achieves the optimal performance with a count of 52, significantly surpassing the second-best PDF model. Compared to the Transformer-based model, SSNet reduces MSE by an average of 16.13% and MAE by 11.07%. When compared to the CNN-based model, SSNet lowers

MSE by 21.89% and MAE by 13.94%. Relative to the MLP-based model, SSNet decreases MSE by 10.88% and MAE by 7.07%.

## 4.2 ABLATION STUDY

**Kernel Weights Sharing.** In order to verify the ability of Skip-MLP and Split-MLP to extract periodic and common features between variables based on shared kernel weights parameters, we conducted verification based on 4 long sequence datasets. For intuitive verification, we only conducted verification based on the simplest network unit, which was divided into 5 groups: 1) the original MLP network as a control; 2) the Skip-MLP network with shared parameters; 3) the Skip-MLP network with independent parameters; 4) the Split-MLP network with shared parameters; 5) the Split-MLP network with independent parameters. The experimental results are shown in Table 3. Experimental results show that Skip-MLP and Split-MLP significantly outperform traditional MLPs

Table 3: Multivariate long-term time series forecasting results for different MLP-based networks with prediction horizons $H = \{96, 192, 336, 720\}$. The best prediction results or the lowest parameter counts are highlighted in **bold red**.

| Models | | MLP | | | Skip-MLP | | | Skip-MLP(I) | | | Split-MLP | | | Split-MLP(I) | | |
|---|---|---|---|---|---|---|---|---|---|---|---|---|---|---|---|---|
| Metric | MSE | Params | Macs | MSE | Params | Macs | MSE | Params | Macs | MSE | Params | Macs | MSE | Params | Macs |
| ETTh1 96 | 0.370 | 3.050M | 2.666G | **0.360** | **0.053M** | **0.154G** | **0.360** | 0.179M | **0.154G** | 0.366 | 0.053M | **0.154G** | 0.364 | 0.179M | **0.154G** |
| ETTh1 192 | 0.410 | 3.097M | 2.707G | 0.396 | **0.099M** | **0.195G** | **0.395** | 0.226M | **0.195G** | 0.405 | 0.010M | **0.195G** | 0.423 | 0.226M | **0.195G** |
| ETTh1 336 | 0.427 | 3.168M | 2.769G | **0.407** | **0.170M** | **0.256G** | **0.407** | 0.297M | **0.256G** | 0.418 | 0.170M | **0.256G** | 0.430 | 0.297M | **0.256G** |
| ETTh1 720 | 0.480 | 3.356M | 2.933G | 0.439 | **0.358M** | **0.420G** | 0.443 | 0.485M | **0.420G** | 0.447 | 0.358M | **0.420G** | 0.455 | 0.484M | **0.420G** |
| ETTm1 96 | 0.310 | 12.059M | 10.54G | **0.285** | **0.049M** | **0.164G** | 0.290 | 0.200M | **0.164G** | 0.306 | 0.071M | **0.164G** | 0.302 | 0.189M | **0.164G** |
| ETTm1 192 | 0.345 | 12.106M | 10.58G | 0.327 | **0.096M** | **0.205G** | 0.324 | 0.247M | **0.205G** | 0.342 | 0.118M | **0.205G** | 0.338 | 0.236M | **0.205G** |
| ETTm1 336 | 0.366 | 12.177M | 10.64G | **0.353** | **0.166M** | **0.267G** | 0.354 | 0.317M | **0.267G** | 0.370 | 0.188M | **0.267G** | 0.368 | 0.306M | **0.267G** |
| ETTm1 720 | 0.424 | 12.364M | 10.80G | **0.406** | **0.354M** | **0.431G** | 0.411 | 0.505M | **0.431G** | 0.423 | 0.376M | **0.431G** | 0.422 | 0.494M | **0.431G** |
| Electricity 96 | 0.136 | 48.09M | 963.9G | 0.129 | **0.135M** | **3.527G** | **0.128** | 2.157M | **3.527G** | 0.165 | 0.143M | **3.527G** | 0.136 | 2.153M | **3.527G** |
| Electricity 192 | 0.151 | 48.14M | 964.8G | **0.148** | **0.182M** | **4.467G** | 0.149 | 2.204M | **4.467G** | 0.150 | 0.190M | **4.467G** | **0.148** | 2.200M | **4.467G** |
| Electricity 336 | 0.166 | 48.21M | 966.2G | 0.161 | **0.252M** | **5.878G** | **0.160** | 2.274M | **5.878G** | 0.165 | 0.260M | **5.878G** | 0.163 | 2.270M | **5.878G** |
| Electricity 720 | 0.206 | 48.40M | 970.0G | **0.203** | **0.440M** | **9.639G** | 0.204 | 2.462M | **9.639G** | 0.205 | 0.448M | **9.639G** | 0.204 | 2.458M | **9.639G** |
| Traffic 96 | 0.395 | 48.09M | 970.6G | **0.362** | **0.135M** | **42.61G** | 0.364 | 2.157M | **42.61G** | 0.385 | 0.143M | **42.61G** | 0.384 | 2.153M | **42.61G** |
| Traffic 192 | 0.412 | 48.14M | 971.6G | 0.380 | **0.182M** | **43.56G** | 0.382 | 2.204M | **43.56G** | 0.398 | 0.190M | **43.56G** | 0.398 | 2.200M | **43.56G** |
| Traffic 336 | 0.419 | 48.21M | 973.0G | 0.390 | **0.252M** | **44.98G** | **0.392** | 2.274M | **44.98G** | 0.407 | 0.260M | **44.98G** | 0.406 | 2.270M | **44.98G** |
| Traffic 720 | 0.461 | 48.40M | 976.8G | 0.428 | **0.440M** | **48.77G** | 0.430 | 2.462M | **48.77G** | 0.442 | 0.448M | **48.77G** | 0.442 | 2.458M | **48.77G** |

under the same hidden layer dimensions in time series forecasting. Moreover, both parameter count and computational cost are drastically reduced (related to the top-1 period). Skip-MLP achieves optimal performance through efficient parameter sharing. As shown in Table 2, a single Skip-MLP with kernel weight sharing matches the performance of state-of-the-art models like PatchTST.

**SkipMLPs and SS-MLP Blocks.** The input layer of SSNet consists of multiple parallel Skip-MLP networks, while the intermediate layer is based on residual connection SS-MLP Blocks. To evaluate the feature extraction capabilities and contributions of these components, we conducted experiments by removing each part separately. "SS Only" denotes the removal of the Skip-MLP networks, retaining only the Auto-correlation Block and SS-MLP Blocks. "Skip Only" indicates the removal of the SS-MLP Blocks.

Experimental results indicate that both the parallel Skip-MLPs in the input layer and the serial SS-MLPs in the intermediate layer are essential. Removing either component significantly degrades the model's performance. This confirms the rationale behind SSNet's architectural design. The results show that the SS-MLPs have a greater impact on overall performance; omitting them leads to a more pronounced decrease in model effectiveness.

Table 4: Multivariate long-term time series forecasting results for different SSNet structures with prediction horizons $H = \{96, 192, 336, 720\}$. The best prediction results are highlighted in **bold red**.

| Methods | SS-Net | | SS-Net (SS Only) | | SS-Net (Skip Only) | |
|---|---|---|---|---|---|---|
| Metric | MSE | MAE | MSE | MAE | MSE | MAE |
| ETTh1 96 | **0.339** | **0.374** | 0.344 | 0.376 | 0.341 | 0.377 |
| ETTh1 192 | **0.359** | **0.386** | 0.366 | 0.390 | 0.360 | 0.390 |
| ETTh1 336 | **0.370** | **0.399** | 0.383 | 0.406 | 0.376 | 0.404 |
| ETTh1 720 | **0.425** | **0.447** | 0.440 | 0.459 | 0.430 | 0.452 |
| ETTm1 96 | **0.277** | **0.340** | 0.280 | 0.342 | 0.297 | 0.353 |
| ETTm1 192 | **0.312** | **0.361** | 0.317 | 0.363 | 0.338 | 0.374 |
| ETTm1 336 | **0.345** | **0.380** | 0.357 | 0.384 | 0.372 | 0.391 |
| ETTm1 720 | **0.404** | **0.409** | 0.419 | 0.416 | 0.424 | 0.416 |
| Electricity 96 | **0.125** | **0.219** | 0.127 | 0.221 | 0.129 | 0.222 |
| Electricity 192 | **0.143** | **0.236** | 0.145 | 0.240 | 0.145 | 0.238 |
| Electricity 336 | **0.158** | **0.252** | 0.160 | 0.256 | 0.162 | 0.257 |
| Electricity 720 | **0.195** | **0.286** | 0.199 | 0.293 | 0.204 | 0.294 |
| Traffic 96 | **0.347** | **0.237** | 0.348 | 0.238 | 0.362 | 0.244 |
| Traffic 192 | **0.370** | **0.248** | 0.373 | 0.249 | 0.381 | 0.252 |
| Traffic 336 | **0.380** | **0.255** | 0.381 | **0.255** | 0.381 | **0.255** |
| Traffic 720 | **0.422** | **0.278** | 0.435 | 0.298 | 0.435 | 0.298 |

### 4.3 ANALYSIS OF EFFICIENCY

The computational complexity of the SSNet model is given by $\mathcal{O}\left(\frac{L}{\bar{S}} + H\right)$ where $L$ and $H$ denote the lengths of the input and output sequences, respectively, and $\bar{S} = \frac{1}{\sum_{i=1}^{K} 1/S_i}$ represents the harmonic mean of the Top-K periods. Transformer-based or CNN-based models generally exhibit higher computational complexity. We conducted experiments on the ETTm2 dataset to evaluate the runtime efficiency and GPU memory usage of various models. The experiments were divided into two groups: (1) we fixed the output length at 96 and varied the input length from 192 to 3072; (2) we fixed the input length at 96 and varied the output length from 192 to 3071.

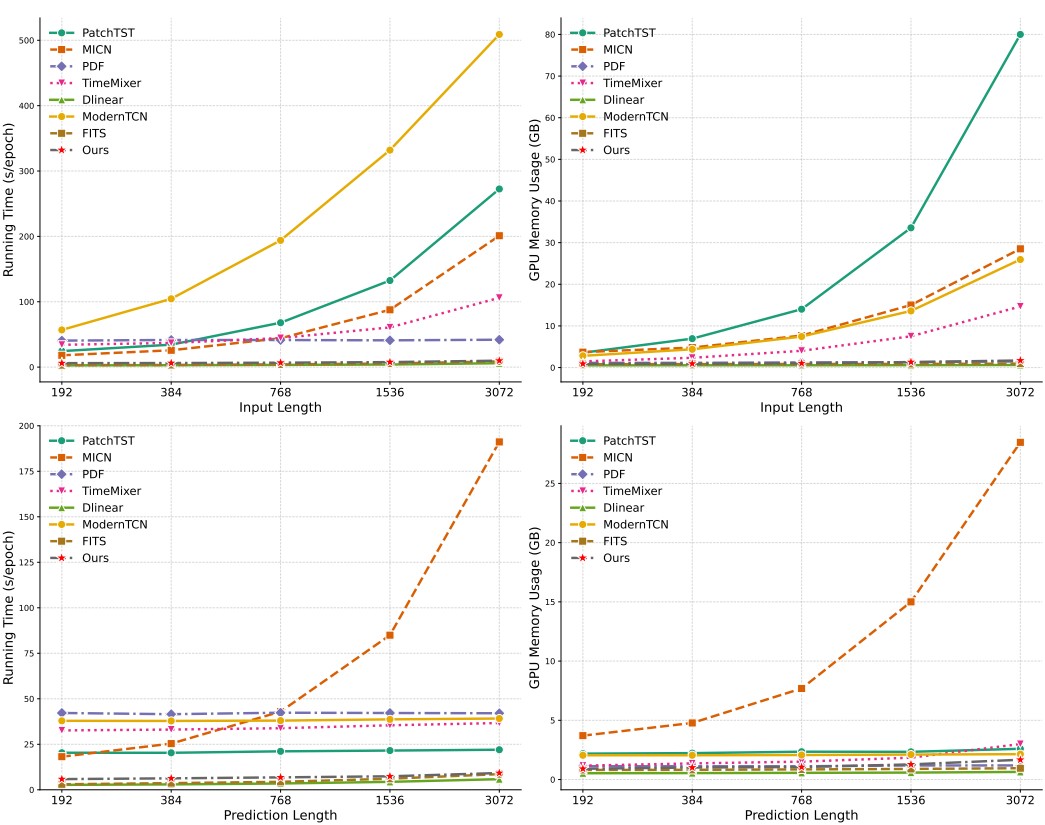

Figure 6: Efficiency and GPU memory usage with varying input lengths and output lengths.

The experimental results indicate that SSNet exhibits exceptional GPU memory usage and runtime efficiency, closely approaching that of the simplest linear model, DLinear. This efficiency enables SSNet to effectively handle large-scale long-sequence forecasting tasks, making it well-suited for applications in industrial production.

## 5 CONCLUSIONS

In this paper, we introduce two innovative architectures, Skip-MLP and Split-MLP, which effectively overcome the limitations of conventional MLPs in capturing periodic information and distinguishing between global and local features. These architectures are distinguished by a reduced parameter count and enhanced computational efficiency. Building on these foundations, we developed the SSNet temporal network, which adaptively identifies significant periodicities through an autocorrelation method, extracts multi-scale periodic features via Skip-MLPs, integrates this multi-scale information, and systematically reduces prediction bias using multiple SS-MLP Blocks with residual connections. SSNet demonstrates significant improvements over existing methods in various long-term time series forecasting tasks, achieving superior prediction performance while utilizing fewer parameters and minimizing runtime.

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

## A  PROOF FOR SKIP-MLP AND SPLIT-MLP

### A.1  PROOF FOR SKIP-MLP

Consider the Skip-MLP model:

$$\mathbf{O_t} = \mathbf{X_t} W_{Skip} + B_{Skip} = [O_1, O_2, \ldots, O_{N_D \times S}] \in \mathbb{R}^{N \times N_D \cdot S}$$

where

$$W_{Skip} = W_{Fully} \odot W_{Mask} \in \mathbb{R}^{N_L \cdot S \times N_D \cdot S}$$

and

$$B_{Skip} \in \mathbb{R}^{1 \times N_D \cdot S}$$

with

$$W_{Mask_{ij}} = \begin{cases} 1 & \text{if } |i - j| \bmod S = 0 \\ 0 & \text{otherwise} \end{cases} \tag{1}$$

This can be expressed as:

$$O_{t_i} = X_{t_i} W_{Fully_i} + B_i = \left[O_i, O_{i+S}, \ldots, O_{i+(N_D-1)S}\right] \in \mathbb{R}^{N \times N_D}$$

$$X_{t_i} = \left[X_i, X_{i+S}, \ldots, X_{i+(N_L-1)S}\right] \in \mathbb{R}^{N \times N_L}$$

$$W_{Fully_i} \in \mathbb{R}^{N_L \times N_D}, B_i \in \mathbb{R}^{1 \times N_D}$$

where $i \in \{1, 2, \ldots, S\}$.

After padding the input $\mathbf{X_t} \in \mathbb{R}^{N \times L}$ to a multiple of $S$, we get

$$\mathbf{X_t} = \text{pad}(\mathbf{X_t}) = \begin{bmatrix} x_{1,1} & x_{1,2} & \cdots & x_{1,N_L \cdot S} \\ x_{2,1} & x_{2,2} & \cdots & x_{2,N_L \cdot S} \\ \vdots & \vdots & \ddots & \vdots \\ x_{N,1} & x_{N,2} & \cdots & x_{N,N_L \cdot S} \end{bmatrix} \in \mathbb{R}^{N \times N_L \cdot S}$$

Consider the permutation matrices $E_{N_L}$ and $E_{N_D}$. The matrix $E_{N_L}$ transforms the permutation $A_L = [a_1, a_2, \ldots, a_{N_L \cdot S}]$ into:

$$A'_L = \left[a_1, a_{S+1}, \ldots, a_{(N_L-1)S+1}, a_2, a_{S+2}, \ldots, a_{(N_L-1)S+2}, \ldots, a_S, a_{2S}, \cdots, a_{N_L \cdot S}\right]$$

which means $A'_L = A_L \cdot E_{N_L}$.

Similarly, $E_{N_D}$ transforms the permutation $A_D = [a_1, a_2, \ldots, a_{N_D \cdot S}]$ into:

$$A'_D = \left[a_1, a_{S+1}, \ldots, a_{(N_D-1)S+1}, a_2, a_{S+2}, \ldots, a_{(N_D-1)S+2}, \ldots, a_S, a_{2S}, \cdots, a_{N_D \cdot S}\right]$$

which means $A'_D = A_D \cdot E_{N_D}$.

Applying these transformations to $\mathbf{X_t}$ and $W_{Skip}$, we get:

$$\mathbf{X'_t} = \mathbf{X_t} \cdot E_{N_L} \tag{2}$$

$$W'_{Skip} = E_{N_L}^T \cdot W_{Skip} \cdot E_{N_D} \tag{3}$$

Let $x_{i,j}$ and $w_{i,j}$ denote the elements at the $i$-th row and $j$-th column of $\mathbf{X_t}$ and $W_{Skip}$ respectively, where $i \in \{1, 2, \ldots, N_L \cdot S\}$ and $j \in \{1, 2, \ldots, N_D \cdot S\}$. From Equation 2, $\mathbf{X'_t}$ can be represented as:

$$\mathbf{X'_t} = \begin{bmatrix} x_{1,1} & x_{1,S+1} & \cdots & x_{1,(N_L-1)S+1} & \cdots & x_{1,S} & x_{1,2S} & \cdots & x_{1,N_L \cdot S} \\ x_{2,1} & x_{2,S+1} & \cdots & x_{2,(N_L-1)S+1} & \cdots & x_{2,S} & x_{2,2S} & \cdots & x_{2,N_L \cdot S} \\ \vdots & \vdots & \ddots & \vdots & \ddots & \vdots & \vdots & \ddots & \vdots \\ x_{N,1} & x_{N,S+1} & \cdots & x_{N,(N_L-1)S+1} & \cdots & x_{N,S} & x_{N,2S} & \cdots & x_{N,N_L \cdot S} \end{bmatrix}$$

From Equation 3, $W'_{Skip}$ can be represented as:

$$W'_{Skip} = \begin{bmatrix} w_{1,1} & w_{1,S+1} & \cdots & w_{1,(N_D-1)S+1} & \cdots & w_{1,S} & w_{1,2S} & \cdots & w_{1,N_D\cdot S} \\ w_{S+1,1} & w_{S+1,S+1} & \cdots & w_{S+1,(N_D-1)S+1} & \cdots & w_{S+1,S} & w_{S+1,2S} & \cdots & w_{S+1,N_D\cdot S} \\ \vdots & \vdots & \ddots & \vdots & \ddots & \vdots & \vdots & \ddots & \vdots \\ w_{(N_L-1)S+1,1} & w_{(N_L-1)S+1,S+1} & \cdots & w_{(N_L-1)S+1,(N_D-1)S+1} & \cdots & w_{(N_L-1)S+1,S} & w_{(N_L-1)S+1,2S} & \cdots & w_{(N_L-1)S+1,N_D\cdot S} \\ \vdots & \vdots & \ddots & \vdots & \ddots & \vdots & \vdots & \ddots & \vdots \\ w_{S,1} & w_{S,S+1} & \cdots & w_{S,(N_D-1)S+1} & \cdots & w_{S,S} & w_{S,2S} & \cdots & w_{S,N_D\cdot S} \\ w_{2S,1} & w_{2S,S+1} & \cdots & w_{2S,(N_D-1)S+1} & \cdots & w_{2S,S} & w_{2S,2S} & \cdots & w_{2S,N_D\cdot S} \\ \vdots & \vdots & \ddots & \vdots & \ddots & \vdots & \vdots & \ddots & \vdots \\ w_{N_L\cdot S,1} & w_{N_L\cdot S,S+1} & \cdots & w_{N_L\cdot S,(N_D-1)S+1} & \cdots & w_{N_L\cdot S,S} & w_{N_L\cdot S,2S} & \cdots & w_{N_L\cdot S,N_D\cdot S} \end{bmatrix}$$

From Equation 1, when $|i - j| \bmod S = 0$, there are non-zero values; otherwise, they are masked to zero. Thus, the matrix $W'_{Skip}$ can be expressed as:

$$W'_{Skip} = \begin{bmatrix} W_{Fully_1} & O & \cdots & O \\ O & W_{Fully_2} & \cdots & O \\ \vdots & \vdots & \ddots & \vdots \\ O & O & \cdots & W_{Fully_S} \end{bmatrix}$$

where

$$W_{Fully_k} = \begin{bmatrix} w_{k,k} & w_{k,S+k} & \cdots & w_{k,(N_D-1)S+k} \\ w_{S+k,k} & w_{S+k,S+k} & \cdots & w_{S+k,(N_D-1)S+k} \\ \vdots & \vdots & \ddots & \vdots \\ w_{(N_L-1)S+k,k} & w_{(N_L-1)S+k,S+k} & \cdots & w_{(N_L-1)S+k,(N_D-1)S+k} \end{bmatrix}, k \in \{1, 2, \ldots, S\}$$

Hence,

$$\mathbf{X'_t} W'_{Skip} = (\mathbf{X_t} E_{N_L})(E_{N_L}^T W_{Skip} E_{N_D}) = \mathbf{X_t}(E_{N_L} E_{N_L}^T) W_{Skip} E_{N_D} = \mathbf{X_t} W_{Skip} E_{N_D}$$

Thus,

$$\begin{aligned} \mathbf{O_t} &= \mathbf{X_t} W_{Skip} + B_{Skip} \\ &= \mathbf{X'_t} W'_{Skip} E_{N_D}^T + B_{Skip} \\ &= [X_{t_1} W_{Fully_1}, X_{t_2} W_{Fully_2}, \ldots, X_{t_S} W_{Fully_S}] E_{N_D}^T + B_{Skip} \end{aligned}$$

By applying the permutation matrix $E_{N_D}$ to $\mathbf{O_t}$, we have:

$$\begin{aligned} \mathbf{O_t} E_{N_D} &= [O_1, O_{1+S}, \ldots, O_{1+(N_D-1)S}, O_2, O_{2+S}, \ldots, O_{2+(N_D-1)S}, \ldots O_S, O_{2S}, \ldots, O_{N_D S}] \\ &= [X_{t_1} W_{Fully_1}, X_{t_2} W_{Fully_2}, \ldots, X_{t_S} W_{Fully_S}] + B_{Skip} E_{N_D} \end{aligned}$$

which proves

$$O_{t_i} = X_{t_i} W_{Fully_i} + B_i = [O_i, O_{i+S}, \ldots, O_{i+(N_D-1)S}] \in \mathbb{R}^{N \times N_D}$$
$$X_{t_i} = [X_i, X_{i+S}, \ldots, X_{i+(N_L-1)S}] \in \mathbb{R}^{N \times N_L}$$
$$W_{Fully_i} \in \mathbb{R}^{N_L \times N_D}, \quad B_i \in \mathbb{R}^{1 \times N_D}, \quad i \in \{1, 2, \ldots, S\}$$

## A.2 PROOF FOR SPLIT-MLP

For the Split-MLP model, let $N_L = N_D = \lceil \frac{L}{S} \rceil$ and $S_D = \lceil \frac{D}{N_D} \rceil$.

$$\mathbf{O_t} = \mathbf{X_t} W_{Split} + B_{Split} = [O_1, O_2, \ldots, O_{N_D \times S_D}] \in \mathbb{R}^{N \times N_D \cdot S_D}$$

where

$$W_{Split} = W_{Fully} \odot W_{Mask} \in \mathbb{R}^{S \times S_D}, B_{Split} \in \mathbb{R}^{1 \times S_D}$$

with

$$W_{Mask_{ij}} = \begin{cases} 1 & \text{if } \lceil \frac{i}{S} \rceil = \lceil \frac{j}{S_D} \rceil \\ 0 & \text{otherwise} \end{cases} \tag{4}$$

Can be expressed as:

$$\mathbf{O_t} = \left[O_{t_1}, O_{t_2}, \cdots, O_{t_{N_D}}\right] \in \mathbb{R}^{N \times N_D \cdot S_D}$$

$$O_{t_i} = X_{t_i} W_{Fully_i} + B_i = [O_i, O_{i+1}, \ldots, O_{i+S_D-1}] \in \mathbb{R}^{N \times S_D}$$

where

$$X_{t_i} = [X_i, X_{i+1}, \ldots, X_{i+S-1}] \in \mathbb{R}^{N \times S}$$

$$W_{Fully_i} \in \mathbb{R}^{S \times S_D}, B_i \in \mathbb{R}^{1 \times S_D}, i \in \{1, 2, \ldots, N_D\}$$

According to Equation 4, it follows that there is a non-zero value if and only if $\left\lceil \frac{i}{S} \right\rceil = \left\lceil \frac{j}{S_D} \right\rceil$, which implies that after splitting the matrix into blocks of size $S \times S_D$, only the diagonal blocks of size $S \times S_D$ have non-zero values, while the rest are zero. Therefore, the weight matrix $W_{Split}$ can be represented as:

$$W_{Split} = \begin{bmatrix} W_{Fully_1} & O & \cdots & O \\ O & W_{Fully_2} & \cdots & O \\ \vdots & \vdots & \ddots & \vdots \\ O & O & \cdots & W_{Fully_S} \end{bmatrix}$$

where:

$$W_{Fully_k} = \begin{bmatrix} w_{(k-1)S+1,(k-1)S_D+1} & w_{(k-1)S+1,(k-1)S_D+2} & \cdots & w_{(k-1)S+1,kS_D} \\ w_{(k-1)S+2,(k-1)S_D+1} & w_{(k-1)S+2,(k-1)S_D+2} & \cdots & w_{(k-1)S+2,kS_D} \\ \vdots & \vdots & \ddots & \vdots \\ w_{kS,(k-1)S_D+1} & w_{kS,(k-1)S_D+2} & \cdots & w_{kS,kS_D} \end{bmatrix} \in \mathbb{R}^{S \times S_D}$$

$$\left\lceil \frac{i}{S} \right\rceil = \left\lceil \frac{j}{S_D} \right\rceil = k, \quad k \in \{1, 2, \ldots, S\}$$

Thus:

$$\mathbf{O_t} = \mathbf{X_t} W_{Split} + B_{Split} = [X_{t_1} W_{Fully_1}, X_{t_2} W_{Fully_2}, \cdots, X_{t_S} W_{Fully_S}] + B_{Split}$$

$$O_{t_i} = X_{t_i} W_{Fully_i} + B_i = [O_i, O_{i+1}, \ldots, O_{i+S_D-1}] \in \mathbb{R}^{N \times S_D}$$

where

$$X_{t_i} = [X_i, X_{i+1}, \ldots, X_{i+S-1}] \in \mathbb{R}^{N \times S}$$

$$W_{Fully_i} \in \mathbb{R}^{S \times S_D}, B_i \in \mathbb{R}^{1 \times S_D}, i \in \{1, 2, \ldots, N_D\}$$

Therefore, the proof is complete.

# B   DETAILED INTRODUCTION OF THE DATASETS

- **ETT (Electricity Transformer Temperature)**: The ETT dataset is collected from a power grid, recording electricity transformer temperatures alongside other features such as load, oil temperature, and ambient conditions. It is divided into four sub-datasets: ETTh1, ETTh2, ETTm1, and ETTm2. The ETTh datasets represent hourly data, while the ETTm datasets record data every 15 minutes. Each sub-dataset spans two years and contains 7 features, including power load, transformer oil temperature, and ambient temperature. This dataset is particularly suited for both short-term and long-term time-series forecasting tasks.

- **Weather**: The Weather dataset contains meteorological data recorded every 10 minutes throughout the year 2020. It includes 21 meteorological indicators, such as air temperature, humidity, wind speed, and pressure. The dataset provides a high-resolution view of weather patterns and is commonly used in environmental forecasting and anomaly detection tasks.

- **Electricity**: The Electricity dataset contains hourly electricity consumption data from 321 clients between 2012 and 2014. This dataset is challenging due to its high variability and non-stationary characteristics, making it a popular choice for evaluating the robustness of forecasting models.

- **Traffic**: The Traffic dataset contains road occupancy rates measured by 862 different sensors on freeways in the San Francisco Bay Area over a two-year period. Data was collected by the California Department of Transportation. This dataset captures dynamic traffic patterns and is widely used in transportation forecasting and optimization research.

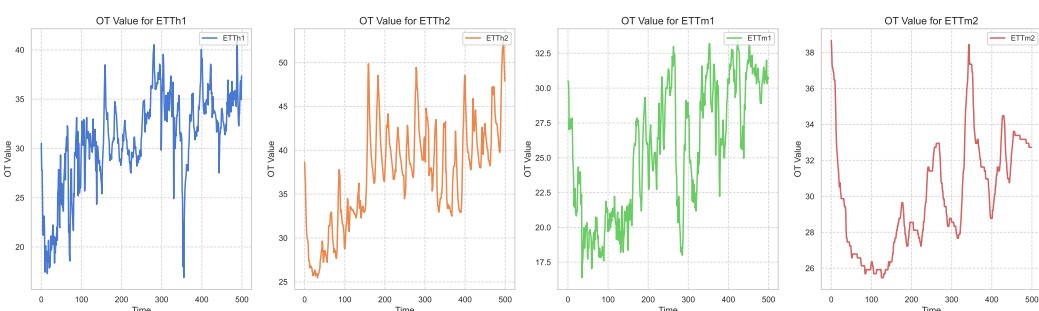

Figure 7: Samples of the ETT dataset.

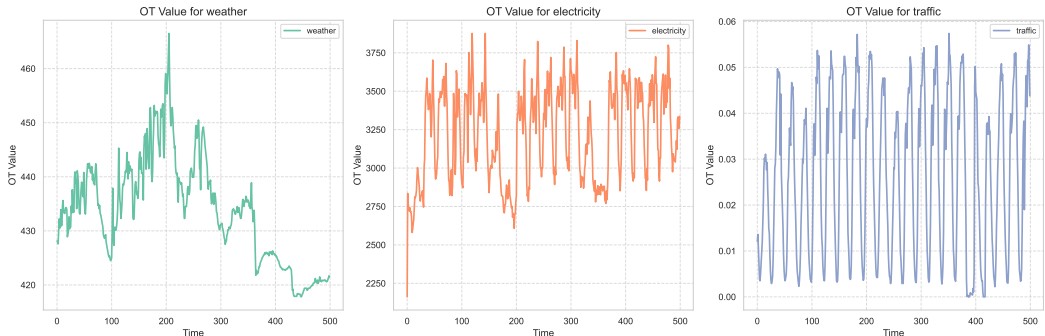

Figure 8: Samples of the Weather, Electricity, and Traffic datasets.

## C  EXPERIMENTS WITH IDENTICAL LOOKBACK WINDOWS

We conducted experiments on four datasets from the ETT dataset, evaluating nine state-of-the-art models, including iTransformer Liu et al. (2023), PDF Dai et al. (2024), PatchTST Nie et al. (2022), ModernTCN Luo & Wang (2024), FITS Xu et al. (2023), Koopa Liu et al. (2024a), CrossGNN Huang et al. (2023), FourierGNN Yi et al. (2024a), and FreTS Yi et al. (2024b), using fixed lookback windows of 96, 336, and 512. The results demonstrate that SSNet consistently outperforms other models in predictive accuracy across different lookback lengths, with performance improving as the sequence length increases. Specifically, for the lookback window of 96, SSNet achieved an average reduction in MSE of 12.36% and an average reduction in MAE of 8.87% compared to other models. For a lookback window of 336, the MSE decreased by an average of 19.56%, and the MAE decreased by 12.16%. With a lookback window of 512, the MSE decreased by 19.76% on average, and the MAE decreased by 12.39%.

Table 5: Experimental Results with a Fixed Lookback Window of 96

| Models | | SSNet | | iTransformer | | PDF | | PatchTST | | ModernTCN | | FITS | | Koopa | | CrossGNN | | FourierGNN | | FreTS | |
|---|---|---|---|---|---|---|---|---|---|---|---|---|---|---|---|---|---|---|---|---|---|
| Metric | | MSE | MAE | MSE | MAE | MSE | MAE | MSE | MAE | MSE | MAE | MSE | MAE | MSE | MAE | MSE | MAE | MSE | MAE | MSE | MAE |
| ETTh1 | 96 | 0.378 | 0.387 | 0.386 | 0.407 | 0.376 | 0.397 | 0.394 | 0.408 | 0.385 | 0.397 | 0.385 | 0.394 | 0.391 | 0.409 | 0.382 | 0.396 | 0.443 | 0.450 | 0.392 | 0.406 |
| | 192 | 0.413 | 0.404 | 0.445 | 0.439 | 0.429 | 0.426 | 0.445 | 0.434 | 0.438 | 0.424 | 0.435 | 0.422 | 0.455 | 0.442 | 0.428 | 0.424 | 0.500 | 0.484 | 0.451 | 0.443 |
| | 336 | 0.435 | 0.421 | 0.485 | 0.460 | 0.464 | 0.441 | 0.484 | 0.451 | 0.444 | 0.428 | 0.474 | 0.440 | 0.490 | 0.457 | 0.471 | 0.442 | 0.554 | 0.517 | 0.503 | 0.471 |
| | 720 | 0.463 | 0.456 | 0.509 | 0.495 | 0.481 | 0.475 | 0.480 | 0.471 | 0.474 | 0.464 | 0.457 | 0.455 | 0.533 | 0.493 | 0.476 | 0.464 | 0.658 | 0.597 | 0.581 | 0.547 |
| | Avg | 0.422 | 0.417 | 0.456 | 0.450 | 0.437 | 0.435 | 0.451 | 0.441 | 0.435 | 0.428 | 0.438 | 0.428 | 0.467 | 0.450 | 0.439 | 0.431 | 0.539 | 0.512 | 0.482 | 0.467 |
| ETTh2 | 96 | 0.228 | 0.300 | 0.304 | 0.352 | 0.294 | 0.342 | 0.294 | 0.343 | 0.281 | 0.335 | 0.294 | 0.341 | 0.307 | 0.356 | 0.289 | 0.341 | 0.429 | 0.457 | 0.313 | 0.364 |
| | 192 | 0.290 | 0.340 | 0.379 | 0.398 | 0.378 | 0.393 | 0.377 | 0.393 | 0.361 | 0.387 | 0.377 | 0.391 | 0.388 | 0.407 | 0.380 | 0.403 | 0.482 | 0.475 | 0.417 | 0.432 |
| | 336 | 0.340 | 0.380 | 0.389 | 0.414 | 0.380 | 0.408 | 0.381 | 0.409 | 0.349 | 0.388 | 0.398 | 0.416 | 0.385 | 0.410 | 0.421 | 0.441 | 0.618 | 0.557 | 0.474 | 0.467 |
| | 720 | 0.410 | 0.431 | 0.415 | 0.437 | 0.414 | 0.435 | 0.412 | 0.433 | 0.432 | 0.444 | 0.412 | 0.432 | 0.425 | 0.441 | 0.433 | 0.455 | 0.866 | 0.690 | 0.772 | 0.615 |
| | Avg | 0.317 | 0.363 | 0.372 | 0.400 | 0.367 | 0.395 | 0.366 | 0.395 | 0.356 | 0.389 | 0.370 | 0.395 | 0.376 | 0.404 | 0.381 | 0.410 | 0.599 | 0.545 | 0.494 | 0.470 |
| ETTm1 | 96 | 0.343 | 0.373 | 0.344 | 0.379 | 0.322 | 0.361 | 0.321 | 0.360 | 0.320 | 0.363 | 0.355 | 0.375 | 0.328 | 0.366 | 0.329 | 0.369 | 0.408 | 0.421 | 0.336 | 0.374 |
| | 192 | 0.374 | 0.390 | 0.386 | 0.398 | 0.359 | 0.380 | 0.365 | 0.382 | 0.369 | 0.384 | 0.392 | 0.393 | 0.372 | 0.389 | 0.370 | 0.391 | 0.432 | 0.436 | 0.383 | 0.402 |
| | 336 | 0.406 | 0.406 | 0.429 | 0.427 | 0.388 | 0.402 | 0.391 | 0.401 | 0.424 | 0.415 | 0.424 | 0.415 | 0.401 | 0.411 | 0.401 | 0.413 | 0.467 | 0.460 | 0.422 | 0.434 |
| | 720 | 0.479 | 0.445 | 0.493 | 0.460 | 0.453 | 0.436 | 0.456 | 0.436 | 0.462 | 0.444 | 0.484 | 0.448 | 0.462 | 0.447 | 0.453 | 0.444 | 0.520 | 0.491 | 0.492 | 0.475 |
| | Avg | 0.400 | 0.403 | 0.413 | 0.416 | 0.380 | 0.395 | 0.383 | 0.395 | 0.388 | 0.401 | 0.414 | 0.408 | 0.390 | 0.403 | 0.388 | 0.404 | 0.457 | 0.452 | 0.408 | 0.421 |
| ETTm2 | 96 | 0.160 | 0.250 | 0.189 | 0.274 | 0.174 | 0.258 | 0.178 | 0.260 | 0.172 | 0.255 | 0.183 | 0.266 | 0.180 | 0.263 | 0.176 | 0.258 | 0.230 | 0.326 | 0.182 | 0.269 |
| | 192 | 0.215 | 0.283 | 0.254 | 0.313 | 0.241 | 0.302 | 0.249 | 0.306 | 0.238 | 0.298 | 0.247 | 0.305 | 0.243 | 0.304 | 0.243 | 0.302 | 0.326 | 0.390 | 0.254 | 0.324 |
| | 336 | 0.260 | 0.313 | 0.317 | 0.352 | 0.302 | 0.341 | 0.303 | 0.341 | 0.302 | 0.339 | 0.308 | 0.343 | 0.301 | 0.341 | 0.302 | 0.338 | 0.385 | 0.426 | 0.336 | 0.376 |
| | 720 | 0.322 | 0.351 | 0.414 | 0.406 | 0.399 | 0.395 | 0.399 | 0.397 | 0.396 | 0.394 | 0.406 | 0.397 | 0.396 | 0.397 | 0.402 | 0.396 | 0.703 | 0.610 | 0.535 | 0.494 |
| | Avg | 0.239 | 0.299 | 0.293 | 0.337 | 0.279 | 0.324 | 0.282 | 0.326 | 0.277 | 0.321 | 0.286 | 0.328 | 0.280 | 0.326 | 0.281 | 0.324 | 0.411 | 0.438 | 0.327 | 0.366 |

Table 6: Experimental Results with a Fixed Lookback Window of 336

| Models | | SSNet | | iTransformer | | PDF | | PatchTST | | ModernTCN | | FITS | | Koopa | | CrossGNN | | FourierGNN | | FreTS | |
|---|---|---|---|---|---|---|---|---|---|---|---|---|---|---|---|---|---|---|---|---|---|
| Metric | | MSE | MAE | MSE | MAE | MSE | MAE | MSE | MAE | MSE | MAE | MSE | MAE | MSE | MAE | MSE | MAE | MSE | MAE | MSE | MAE |
| ETTh1 | 96 | 0.343 | 0.375 | 0.406 | 0.422 | 0.365 | 0.391 | 0.375 | 0.399 | 0.369 | 0.394 | 0.373 | 0.395 | 0.389 | 0.415 | 0.369 | 0.392 | 0.478 | 0.472 | 0.412 | 0.432 |
| | 192 | 0.364 | 0.387 | 0.449 | 0.448 | 0.402 | 0.413 | 0.414 | 0.421 | 0.406 | 0.414 | 0.406 | 0.414 | 0.438 | 0.442 | 0.416 | 0.426 | 0.511 | 0.495 | 0.460 | 0.465 |
| | 336 | 0.376 | 0.400 | 0.449 | 0.455 | 0.411 | 0.423 | 0.431 | 0.436 | 0.427 | 0.425 | 0.427 | 0.425 | 0.455 | 0.455 | 0.448 | 0.448 | 0.563 | 0.538 | 0.483 | 0.475 |
| | 720 | 0.429 | 0.449 | 0.534 | 0.524 | 0.444 | 0.460 | 0.450 | 0.466 | 0.450 | 0.461 | 0.421 | 0.442 | 0.473 | 0.480 | 0.460 | 0.467 | 0.685 | 0.633 | 0.629 | 0.563 |
| | Avg | 0.378 | 0.403 | 0.460 | 0.462 | 0.406 | 0.422 | 0.417 | 0.431 | 0.404 | 0.421 | 0.407 | 0.419 | 0.439 | 0.448 | 0.423 | 0.433 | 0.559 | 0.534 | 0.496 | 0.484 |
| ETTh2 | 96 | 0.219 | 0.297 | 0.305 | 0.361 | 0.275 | 0.337 | 0.274 | 0.336 | 0.264 | 0.333 | 0.277 | 0.339 | 0.301 | 0.360 | 0.285 | 0.341 | 0.382 | 0.429 | 0.304 | 0.357 |
| | 192 | 0.267 | 0.334 | 0.389 | 0.412 | 0.339 | 0.382 | 0.339 | 0.379 | 0.318 | 0.373 | 0.337 | 0.377 | 0.358 | 0.399 | 0.359 | 0.393 | 0.499 | 0.498 | 0.383 | 0.410 |
| | 336 | 0.306 | 0.367 | 0.383 | 0.415 | 0.331 | 0.387 | 0.330 | 0.380 | 0.314 | 0.376 | 0.342 | 0.388 | 0.349 | 0.399 | 0.368 | 0.411 | 0.550 | 0.530 | 0.465 | 0.466 |
| | 720 | 0.379 | 0.421 | 0.412 | 0.443 | 0.383 | 0.423 | 0.379 | 0.422 | 0.415 | 0.445 | 0.379 | 0.419 | 0.418 | 0.444 | 0.480 | 0.526 | 0.932 | 0.711 | 1.067 | 0.711 |
| | Avg | 0.293 | 0.355 | 0.372 | 0.408 | 0.332 | 0.382 | 0.331 | 0.379 | 0.328 | 0.382 | 0.334 | 0.381 | 0.357 | 0.400 | 0.373 | 0.418 | 0.591 | 0.542 | 0.555 | 0.486 |
| ETTm1 | 96 | 0.273 | 0.336 | 0.312 | 0.364 | 0.284 | 0.336 | 0.292 | 0.343 | 0.297 | 0.348 | 0.304 | 0.345 | 0.304 | 0.356 | 0.297 | 0.342 | 0.354 | 0.404 | 0.332 | 0.369 |
| | 192 | 0.311 | 0.360 | 0.354 | 0.389 | 0.319 | 0.361 | 0.331 | 0.369 | 0.346 | 0.376 | 0.337 | 0.365 | 0.341 | 0.379 | 0.337 | 0.366 | 0.395 | 0.428 | 0.362 | 0.386 |
| | 336 | 0.348 | 0.381 | 0.383 | 0.405 | 0.356 | 0.385 | 0.365 | 0.392 | 0.376 | 0.395 | 0.372 | 0.385 | 0.377 | 0.403 | 0.367 | 0.384 | 0.439 | 0.461 | 0.400 | 0.413 |
| | 720 | 0.407 | 0.413 | 0.446 | 0.441 | 0.408 | 0.416 | 0.417 | 0.423 | 0.429 | 0.423 | 0.427 | 0.416 | 0.435 | 0.432 | 0.420 | 0.414 | 0.490 | 0.484 | 0.456 | 0.449 |
| | Avg | 0.335 | 0.373 | 0.374 | 0.400 | 0.342 | 0.374 | 0.351 | 0.382 | 0.362 | 0.385 | 0.360 | 0.378 | 0.364 | 0.392 | 0.355 | 0.376 | 0.419 | 0.444 | 0.387 | 0.404 |
| ETTm2 | 96 | 0.140 | 0.235 | 0.172 | 0.265 | 0.165 | 0.255 | 0.165 | 0.255 | 0.170 | 0.255 | 0.166 | 0.255 | 0.177 | 0.263 | 0.162 | 0.249 | 0.226 | 0.327 | 0.185 | 0.266 |
| | 192 | 0.180 | 0.264 | 0.243 | 0.314 | 0.220 | 0.292 | 0.221 | 0.292 | 0.228 | 0.299 | 0.221 | 0.292 | 0.241 | 0.308 | 0.221 | 0.294 | 0.284 | 0.360 | 0.185 | 0.266 |
| | 336 | 0.219 | 0.291 | 0.287 | 0.341 | 0.274 | 0.328 | 0.278 | 0.329 | 0.290 | 0.338 | 0.275 | 0.327 | 0.301 | 0.347 | 0.275 | 0.331 | 0.354 | 0.403 | 0.263 | 0.315 |
| | 720 | 0.280 | 0.332 | 0.373 | 0.393 | 0.363 | 0.383 | 0.367 | 0.385 | 0.375 | 0.392 | 0.366 | 0.382 | 0.376 | 0.395 | 0.369 | 0.388 | 0.569 | 0.529 | 0.361 | 0.367 |
| | Avg | 0.205 | 0.280 | 0.269 | 0.328 | 0.256 | 0.315 | 0.258 | 0.315 | 0.266 | 0.321 | 0.257 | 0.314 | 0.274 | 0.328 | 0.257 | 0.316 | 0.358 | 0.405 | 0.446 | 0.439 |

Table 7: Experimental Results with a Fixed Lookback Window of 512

| Models | | SSNet | | iTransformer | | PDF | | PatchTST | | ModernTCN | | FITS | | Koopa | | CrossGNN | | FourierGNN | | FreTS | |
|---|---|---|---|---|---|---|---|---|---|---|---|---|---|---|---|---|---|---|---|---|---|
| Metric | | MSE | MAE | MSE | MAE | MSE | MAE | MSE | MAE | MSE | MAE | MSE | MAE | MSE | MAE | MSE | MAE | MSE | MAE | MSE | MAE |
| ETTh1 | 96 | 0.339 | 0.374 | 0.400 | 0.424 | 0.359 | 0.391 | 0.370 | 0.400 | 0.367 | 0.396 | 0.371 | 0.396 | 0.387 | 0.417 | 0.370 | 0.397 | 0.482 | 0.477 | 0.428 | 0.443 |
| | 192 | 0.359 | 0.386 | 0.426 | 0.443 | 0.392 | 0.414 | 0.413 | 0.429 | 0.403 | 0.415 | 0.405 | 0.415 | 0.423 | 0.441 | 0.429 | 0.437 | 0.508 | 0.496 | 0.463 | 0.470 |
| | 336 | 0.370 | 0.398 | 0.431 | 0.452 | 0.409 | 0.431 | 0.422 | 0.440 | 0.395 | 0.416 | 0.418 | 0.427 | 0.441 | 0.453 | 0.425 | 0.433 | 0.532 | 0.515 | 0.493 | 0.484 |
| | 720 | 0.427 | 0.449 | 0.556 | 0.537 | 0.472 | 0.482 | 0.447 | 0.468 | 0.461 | 0.470 | 0.421 | 0.444 | 0.544 | 0.519 | 0.458 | 0.474 | 0.659 | 0.615 | 0.586 | 0.542 |
| | Avg | 0.374 | 0.402 | 0.453 | 0.464 | 0.408 | 0.429 | 0.413 | 0.434 | 0.407 | 0.424 | 0.404 | 0.421 | 0.449 | 0.458 | 0.421 | 0.435 | 0.545 | 0.526 | 0.492 | 0.485 |
| ETTh2 | 96 | 0.216 | 0.298 | 0.306 | 0.363 | 0.272 | 0.337 | 0.274 | 0.337 | 0.256 | 0.329 | 0.272 | 0.337 | 0.308 | 0.364 | 0.277 | 0.346 | 0.384 | 0.427 | 0.313 | 0.368 |
| | 192 | 0.262 | 0.332 | 0.379 | 0.408 | 0.333 | 0.377 | 0.340 | 0.381 | 0.310 | 0.369 | 0.330 | 0.375 | 0.358 | 0.401 | 0.343 | 0.396 | 0.441 | 0.466 | 0.390 | 0.421 |
| | 336 | 0.300 | 0.364 | 0.391 | 0.423 | 0.326 | 0.383 | 0.329 | 0.384 | 0.315 | 0.377 | 0.339 | 0.387 | 0.360 | 0.408 | 0.382 | 0.411 | 0.549 | 0.540 | 0.494 | 0.487 |
| | 720 | 0.372 | 0.419 | 0.434 | 0.459 | 0.394 | 0.436 | 0.380 | 0.423 | 0.411 | 0.444 | 0.372 | 0.418 | 0.453 | 0.471 | 0.428 | 0.471 | 0.702 | 0.615 | 1.273 | 0.774 |
| | Avg | 0.287 | 0.353 | 0.378 | 0.413 | 0.331 | 0.383 | 0.331 | 0.381 | 0.323 | 0.380 | 0.328 | 0.379 | 0.370 | 0.411 | 0.357 | 0.406 | 0.519 | 0.512 | 0.618 | 0.512 |
| ETTm1 | 96 | 0.277 | 0.340 | 0.315 | 0.367 | 0.281 | 0.337 | 0.290 | 0.344 | 0.309 | 0.355 | 0.307 | 0.349 | 0.314 | 0.364 | 0.304 | 0.348 | 0.363 | 0.406 | 0.346 | 0.379 |
| | 192 | 0.312 | 0.361 | 0.351 | 0.387 | 0.326 | 0.361 | 0.334 | 0.371 | 0.346 | 0.376 | 0.334 | 0.367 | 0.347 | 0.384 | 0.345 | 0.371 | 0.402 | 0.425 | 0.381 | 0.408 |
| | 336 | 0.345 | 0.380 | 0.382 | 0.409 | 0.352 | 0.386 | 0.369 | 0.392 | 0.377 | 0.395 | 0.368 | 0.384 | 0.379 | 0.407 | 0.375 | 0.392 | 0.439 | 0.448 | 0.405 | 0.419 |
| | 720 | 0.404 | 0.410 | 0.442 | 0.441 | 0.410 | 0.417 | 0.416 | 0.420 | 0.428 | 0.420 | 0.421 | 0.413 | 0.438 | 0.432 | 0.423 | 0.417 | 0.490 | 0.478 | 0.465 | 0.453 |
| | Avg | 0.335 | 0.373 | 0.373 | 0.401 | 0.342 | 0.377 | 0.352 | 0.382 | 0.365 | 0.386 | 0.358 | 0.378 | 0.369 | 0.397 | 0.362 | 0.382 | 0.423 | 0.439 | 0.399 | 0.415 |
| ETTm2 | 96 | 0.139 | 0.233 | 0.179 | 0.273 | 0.162 | 0.252 | 0.166 | 0.256 | 0.171 | 0.262 | 0.165 | 0.254 | 0.187 | 0.275 | 0.161 | 0.251 | 0.240 | 0.338 | 0.208 | 0.272 |
| | 192 | 0.181 | 0.263 | 0.245 | 0.315 | 0.224 | 0.297 | 0.223 | 0.296 | 0.229 | 0.303 | 0.219 | 0.291 | 0.244 | 0.314 | 0.223 | 0.296 | 0.330 | 0.399 | 0.297 | 0.323 |
| | 336 | 0.219 | 0.291 | 0.290 | 0.344 | 0.267 | 0.328 | 0.274 | 0.329 | 0.293 | 0.344 | 0.272 | 0.326 | 0.300 | 0.347 | 0.283 | 0.338 | 0.393 | 0.436 | 0.330 | 0.361 |
| | 720 | 0.279 | 0.332 | 0.369 | 0.393 | 0.346 | 0.377 | 0.362 | 0.385 | 0.386 | 0.401 | 0.357 | 0.380 | 0.372 | 0.397 | 0.367 | 0.387 | 0.755 | 0.629 | 0.391 | 0.414 |
| | Avg | 0.204 | 0.280 | 0.271 | 0.331 | 0.250 | 0.313 | 0.256 | 0.317 | 0.270 | 0.328 | 0.254 | 0.313 | 0.276 | 0.334 | 0.259 | 0.318 | 0.430 | 0.450 | 0.306 | 0.342 |

