# OpenReview forum: "SSNet: Skip and Split MLP Network for Long-Term  Series Forecasting"
_ICLR.cc/2025/Conference — Submitted to ICLR 2025_

### Official Review · Reviewer_iKJa · 2024-10-24

**Soundness:** 4
**Presentation:** 4
**Contribution:** 4
**Rating:** 10
**Confidence:** 5

**Summary:**

In this paper, the authors propose a novel network structure, the Skip and Split MLP Network (SSNet), which integrates Skip-MLP and Split-MLP components. The SSNet model outperforms general MLP-based, CNN-based, and Transformer-based models in terms of parameter efficiency, computation time, and accuracy. The authors clearly explain the decomposition and prediction challenges in time series forecasting, providing detailed mathematical explanations for Skip-MLP and Split-MLP calculations.

The paper introduces the SS-MLP block, consisting of two Skip-MLP layers and two Split-MLP layers. In the SSNet section, they describe the structure of the Auto-correlation Block, which is based on periodicity and strength, and then present the overall SSNet architecture, including the Auto-correlation Block, Skip-MLPs, and SS-MLPs.

The SSNet model is tested on seven datasets and compared with other MLP-based methods (TimeMixer, DLinear), Transformer-based methods (PatchTST, PDF, FEDformer), and CNN-based methods (FiLM, TimesNet, MICN). SSNet consistently ranks as the best or second-best in terms of MSE and MAE. Additionally, the model demonstrates significantly lower running times and GPU/memory usage compared to other methods.

**Strengths:**

1. The inclusion of clear visuals such as Figures 1, 3, 4, and 5 significantly aids in understanding the components of the model, including Skip-MLP, Split-MLP, and the overall structure of SSNet, SS-MLP Block, and the Auto-correlation Block.

2. The concise mathematical derivation of Skip-MLP and Split-MLP helps readers grasp the underlying mathematical framework of the proposed model.

3. The model was tested on multiple large-scale datasets and compared to several well-known deep learning models, offering substantial evidence of the model's strengths and performance.

4. Sufficient data on computational time and GPU usage is provided, demonstrating that the model achieves improved accuracy without requiring additional time or resources, making it highly efficient.

**Weaknesses:**

1. Please explain the reasoning behind this specific number of layers and whether they experimented with different configurations

2. Offer further explanations regarding the structure of Figure 5, provide a step-by-step explanation of the data flow through the SSNet architecture, including how the residual connections are incorporated.

3. Describe the specific type of linear projection used (e.g., fully connected layer) and explain why this particular approach was chosen for the output layer.

4. Provide a brief description of each dataset's domain and characteristics, and suggest including a sample visualization of the time series data from one or two representative datasets.

**Questions:**

I have just one question: Would changing the order of the Skip-MLP and Split-MLP in the SS-MLP block, or adding additional hidden layers, have any impact on the MSE or MAE?

---

> ### Author Response · Authors · 2024-11-19
> **Response to Reviewer iKJa's Comment**
>
> Thank you for your valuable comments and insightful feedback on our submission. We appreciate the time and effort you invested in reviewing our work, as well as the opportunity to clarify and improve our paper based on your suggestions. Below, we address each identified weakness and question in detail:
>
> ### **Weaknesses**
>
> **W1: Explanation of the Number of Layers**
>
> We understand that your concern is why the SS-MLP is designed with  **2K−1** layers instead of other configurations. This choice is primarily based on our observation that using smaller periods in the SS-MLP for the input and output layers achieves better forecasting performance. To fully exploit all periodic information, it is reasonable to process periods sequentially from small to large and then back from large to small. We experimented with  **K**-layer configurations in either a large-to-small or small-to-large direction, but neither performed as well as the current  design. To avoid introducing additional hyperparameters, we set the number of layers to  **2K−1**, which allows the entire network structure to be controlled by a single  **K** value. This makes the architecture both elegant and manageable.
>
> We will provide an explanation for this question in the revised version of the paper.
>
> **W2: Explanation of Figure 5 and Model Architecture**
>
> Thank you for pointing out the need for further explanation. We will revise the paper to include a step-by-step explanation of the data flow through the SSNet architecture. This will provide more insight into how the residual connections are integrated and how information propagates through the model. We aim to improve the clarity of the model’s structure to help readers better understand its design and functionality.
>
> **W3: Type of Linear Projection Used**
>
> The Skip-MLP and Split-MLP components serve to extract and fuse periodic or local features within the model. For the final output layer, we opted for a fully connected layer (FC layer) to effectively integrate the diverse features extracted by the preceding layers. This choice allows for better prediction results by combining all the extracted features in a coherent manner. The FC layer is particularly well-suited for this task, as it can handle complex feature fusion in a flexible and efficient manner. We will provide an explanation for the choice of a fully connected (FC) layer in the output layer in the revised version of the paper.
>
> **W4: Description of Datasets**
>
> Thank you for the valuable suggestion. We will add a concise description of each dataset's domain and characteristics to provide readers with a clearer understanding of their properties and potential influence on model performance. Furthermore, we plan to include sample visualizations from one or two representative datasets to illustrate the nature of the time series data and its relevance to the forecasting tasks.
>
> ------
>
> ### **Questions**
>
> **Response to Question: Impact of Changing the Order of Skip-MLP and Split-MLP**
>
> Yes, we have experimented with changing the order of Skip-MLP and Split-MLP, which led to an increase in the model's MSE and MAE. We found that the Skip-MLP in the first layer of the SS-MLP block has the greatest impact and cannot be replaced by Split-MLP, as this causes the most significant performance degradation. Similarly, the second layer also cannot be replaced with Skip-MLP. The overall performance ranking is as follows: **Skip+Split > Skip+Skip > Split+Skip > Split+Split**. The **Skip-MLP+Split-MLP** combination is the best configuration we have tried so far.
>
> We have not yet explored the combination with a fully connected layer in other configurations, but intuitively, this could be a promising direction for future work, and we will explore it in our future research.
>
> ------
>
> We hope our responses have appropriately addressed your concerns and provided meaningful insights. Thank you for your thoughtful feedback and for acknowledging our efforts, which encourages us to further enhance our work.

---

> > ### Comment · Reviewer_iKJa · 2024-11-22
> >
> > Thank you for your response. I believe my questions have been resolved.

---

### Official Review · Reviewer_Ko6d · 2024-11-04

**Soundness:** 2
**Presentation:** 2
**Contribution:** 2
**Rating:** 1
**Confidence:** 5

**Summary:**

This paper proposes a novel architecture with Skip-MLP and Split-MLP components that effectively captures periodic and temporal relationships while maintaining computational efficiency. Through extensive evaluation of real-world datasets, this paper demonstrates that SSNet outperforms several models with fewer parameters, with even a single Skip-MLP unit achieving comparable performance to complex models like PatchTST.

**Strengths:**

1. this paper studies a popular problem, i.e., time series forecasting.

2. this paper revised MLP architectures and applied them for forecasting, which seems interesting.

**Weaknesses:**

1. This paper is not well-motivated. This paper mentioned that "MLP-based networks offer better computational efficiency but struggle
to effectively model periodic and temporal relationships, which are essential for accurate time series forecasting". However, several models that utilized frequency-domain MLPs can effectively capture the frequency components, such as [1-2]. These methods can better capture the periodic and temporal patterns.

2. The experimental results are less convincing. The experiments are lack of comparisons with other MLP time series forecasting models and SOTA transformer models [1-3]. Thus, the experimental results are not enough.


[1] FITS: Modeling Time Series with 10k Parameters.

[2] Frequency-domain MLPs are More Effective Learners in Time Series Forecasting.

[3] iTransformer: Inverted Transformers Are Effective for Time Series Forecasting.

**Questions:**

Can the author revise or improve the motivations or better explain it?

Can more experiments be added?

---

> ### Author Response · Authors · 2024-11-19
> **Response to Reviewer Ko6d's Comment**
>
> Thank you for your insightful comments and valuable suggestions. We sincerely thank you for the time and effort you have devoted to reviewing our work. Below we address each of your comments and concerns:
>
> ### **Weaknesses**
>
> **W1: Lack of Accurate Motivation**
>
> Thank you for pointing out the inaccuracy in our description regarding the limitations of MLP-based models. You are correct that several frequency-domain MLP models, such as those mentioned in [1, 2], have demonstrated the ability to effectively capture periodic and temporal patterns. Our intention was to highlight that, compared to traditional MLP architectures, Skip-MLP and Split-MLP offer a more natural and efficient way to model periodic relationships in time series data, leveraging their architectural designs to achieve both higher computational efficiency and improved forecasting performance. We will revise the motivation section to reflect this clarification and ensure greater precision in our arguments.
>
> **W2: Insufficient Experimental Comparisons**
>
> We sincerely appreciate the models you have highlighted, including FITS, frequency-domain MLPs, and iTransformer [1–3]. To address this concern, we will conduct additional experiments to compare SSNet with these models as comprehensively as possible. By including these comparisons, we aim to provide a more thorough evaluation of our model's performance and enhance the credibility of the experimental results. These updates will be reflected in the revised manuscript.
>
> ------
>
> ### **Questions**
>
> **Q1:Can the authors revise or improve the motivations?**
>
> As mentioned above, we will refine the motivation section to provide a more accurate and compelling explanation, incorporating the contributions of existing frequency-domain MLP models while emphasizing the unique advantages of Skip-MLP and Split-MLP.
>
> **Q2:Can more experiments be added?**
>
> Yes, we are actively working on adding new experiments to compare SSNet with additional MLP and state-of-the-art transformer models. These experiments will be conducted to provide a more comprehensive assessment of SSNet’s performance across diverse baselines.  Some experimental results have been included in the response to [Reviewer z6FM](https://openreview.net/forum?id=CgRkPuhTGm&noteId=Zo6sgieOVl#:~:text=%E2%89%A1-,Replying%20to%20Response%20to%20Reviewer%20z6FM%27s%20Comments,-Experiments%20Part%201). The relevant findings will also be reflected in the revised version of the paper.
>
> We greatly appreciate your constructive feedback and believe these improvements will significantly enhance the quality and clarity of the paper. Thank you for helping us refine our work.

---

> > ### Comment · Reviewer_Ko6d · 2024-12-02
> >
> > Thanks for your response. My questions still remain:
> >
> > 1. Directly mentioning Skip/Split MLPs are better for periodic modelling does not make sense so much. Why they are better in modelling periodicity? I think Frequency learning methods are directly designed for the frequency components, while periodicity is one kind of frequency components, which is also good. So what is the necessity of these MLPs?
> >
> > 2. As mentioned by Reviewer z6FM, I am not very convinced by the additional experiments. The ignorance of many related works also shows that the authors do not have comprehensive understandings to this domain.
> >
> > In summary, I will keep my score.

---

### Official Review · Reviewer_z6FM · 2024-11-04

**Soundness:** 2
**Presentation:** 3
**Contribution:** 2
**Rating:** 1
**Confidence:** 5

**Summary:**

This paper focuses on long-range time series forecasting, and introduces a novel MLP-based network, comprising Skip-MLP and Split-MLP, combined into SSNet. The author conducted experiments on 7 datasets to compare the performance of SSNet with past methods.

**Strengths:**

1. The innovativeness of the model is good.

2. The overall presentation of the article is very clear.

**Weaknesses:**

1. The experimental setup has some issues. The article used different look-back lengths for the models being compared, which is evidently unreasonable as the input length variable was not controlled. Although it must be acknowledged that different models may exhibit varying performance at different look-back lengths, a reasonable approach would be to evaluate all models using both long and short look-back lengths to compare their performance. Therefore, the results under the current setup are difficult to be convincing.

2. The author seems to have a bias towards the models being compared. I noticed that the author focused on TimeMixer and PDF but did not pay attention to contemporaneous models like iTransformer[1], FITS[2], ModernTCN[3], or even earlier methods such as Koopa[4], CrossGNN[5], FourierGNN[6], WITRAN[7], and Basisformer[8]. The author should comprehensively compare all of them, as I believe this would enhance the quality of the article.

[1] Liu, Y., Hu, T., Zhang, H., Wu, H., Wang, S., Ma, L., & Long, M. (2024). iTransformer: Inverted Transformers Are Effective for Time Series Forecasting. In The Twelfth International Conference on Learning Representations.

[2] Xu, Z., Zeng, A., & Xu, Q. (2024). FITS: Modeling Time Series with $10 k $ Parameters. In The Twelfth International Conference on Learning Representations.

[3] Luo, D., & Wang, X. (2024). Moderntcn: A modern pure convolution structure for general time series analysis. In The Twelfth International Conference on Learning Representations.

[4] Liu, Y., Li, C., Wang, J., & Long, M. (2024). Koopa: Learning non-stationary time series dynamics with koopman predictors. In Thirty-seventh Conference on Neural Information Processing Systems.

[5] Huang, Q., Shen, L., Zhang, R., Ding, S., Wang, B., Zhou, Z., & Wang, Y. CrossGNN: Confronting Noisy Multivariate Time Series Via Cross Interaction Refinement. In Thirty-seventh Conference on Neural Information Processing Systems.

[6] Yi, K., Zhang, Q., Fan, W., He, H., Hu, L., Wang, P., ... & Niu, Z. FourierGNN: Rethinking Multivariate Time Series Forecasting from a Pure Graph Perspective. In Thirty-seventh Conference on Neural Information Processing Systems.

[7] Jia, Y., Lin, Y., Hao, X., Lin, Y., Guo, S., & Wan, H. (2023). WITRAN: Water-wave Information Transmission and Recurrent Acceleration Network for Long-range Time Series Forecasting. In Thirty-seventh Conference on Neural Information Processing Systems.

[8] Ni, Z., Yu, H., Liu, S., Li, J., & Lin, W. (2023). BasisFormer: Attention-based Time Series Forecasting with Learnable and Interpretable Basis. In Thirty-seventh Conference on Neural Information Processing Systems.

3. It seems that the author has not described the details of the parameter search space for the compared baselines. Has the author validated the baseline methods by searching for the best parameters on the validation set? If this has been done, please disclose the results of the parameter search. If this work has not been carried out, it would not provide strong evidence that SSNet outperforms the compared methods. Because the experimental platform can also affect the accuracy of model training, in other words, the optimal parameters on different platforms should be different. To make a fair comparison of their performance, a sufficiently comprehensive parameter search would be conducted on the same platform for all methods within the same search space to ensure they all achieve the best performance. Otherwise, it is difficult to eliminate the significant impact of the experimental platform and parameter selection on the experimental conclusions. Therefore, the experimental conclusions presented in this paper are difficult to be convincing.

**Questions:**

1. Can the author provide details about the search space and make them public?

---

> ### Author Response · Authors · 2024-11-19
> **Response to Reviewer z6FM's Comments**
>
> Thank you very much for your insightful and constructive feedback. We appreciate your attention to detail and your valuable suggestions, which will help us improve the quality of our work. Below, we address your concerns and the proposed revisions in our paper:
>
> ### **Weaknesses**
>
> **W1: Different look-back lengths for the models being compared.**
>
> Thank you for pointing out the importance of controlling the input length when comparing models. We completely agree that evaluating models with consistent look-back lengths is a fair and practical approach. In response, we have conducted additional experiments where all models are evaluated using both long and short look-back lengths. The complete experimental results will be included at the end of this review and reflected in the revised version of the paper.
>
> **W2: Comparison of SSNet with more contemporaneous models.**
>
> We sincerely appreciate the detailed list of additional models for comparison. In the revised paper, we will incorporate as many of these models as possible into our experiments. By doing so, we aim to enhance the comprehensiveness and persuasiveness of our experimental results.
>
> **W3: Details of parameter search space for baseline models.**
>
> For the baseline models, we primarily followed the parameter configurations provided in their original papers. In most cases, we successfully reproduced results that matched or were close to those reported. For models where optimal results or scripts were unavailable, we referenced their reported results while ensuring alignment with comparable baselines, such as PatchTST. This approach aligns with practices in many previous studies.
>
> That said, we acknowledge your concern that this method may introduce some limitations. To address this, we have conducted new experiments on a unified platform, ensuring consistent parameter settings across all models, including SSNet. By evaluating all models under identical conditions, we mitigate the potential impact of platform-specific and parameter-related variations on our conclusions.
>
> ------
>
> ### **Questions**
>
> **Q1: Can the author provide details about the search space and make them public?**
>
> As mentioned in response to Weakness 3, we primarily relied on the optimal configurations reported in the original papers to ensure comparability. To promote transparency, we have made all experimental scripts, including parameter details, available in the supplementary materials. In the new experiments with unified input lengths, we used identical parameter settings across all models without fine-tuning, ensuring fair and consistent evaluation for all methods.
>
> We hope that these additional experiments and clarifications address your concerns. Thank you for your thoughtful feedback, which has been instrumental in improving the quality of our work.

---

> > ### Author Response · Authors · 2024-11-19
> > **Experiments Part 1 of 4**
> >
> > **Experiments focusing on a lookback window of 96**
> >
> > |           |      |   SSNet   |           | iTransformer |       | ModernTCN |           | FITS  |       | Koopa |       | CrossGNN  |       | FourierGNN |       | FreTS |       |
> > | :-------- | ---- | :-------: | :-------: | :----------: | :---: | :-------: | :-------: | :---: | :---: | :---: | :---: | :-------: | :---: | :--------: | :---: | ----- | ----- |
> > |           |      |    MSE    |    MAE    |     MSE      |  MAE  |    MSE    |    MAE    |  MSE  |  MAE  |  MSE  |  MAE  |    MSE    |  MAE  |    MSE     |  MAE  | MSE   | MAE   |
> > | **ETTh1** | 96   |   0.378   |   0.387   |    0.386     | 0.407 |   0.385   |   0.397   | 0.385 | 0.394 | 0.391 | 0.409 |   0.382   | 0.396 |   0.443    | 0.450 | 0.392 | 0.406 |
> > |           | 192  |   0.413   |   0.404   |    0.445     | 0.439 |   0.438   |   0.424   | 0.435 | 0.422 | 0.455 | 0.442 |   0.428   | 0.424 |   0.500    | 0.484 | 0.451 | 0.443 |
> > |           | 336  |   0.435   |   0.421   |    0.485     | 0.460 |   0.444   |   0.428   | 0.474 | 0.440 | 0.490 | 0.457 |   0.471   | 0.442 |   0.554    | 0.517 | 0.503 | 0.471 |
> > |           | 720  |   0.463   |   0.456   |    0.509     | 0.495 |   0.474   |   0.464   | 0.457 | 0.455 | 0.533 | 0.493 |   0.476   | 0.464 |   0.658    | 0.597 | 0.581 | 0.547 |
> > |           | avg  | **0.422** | **0.417** |    0.456     | 0.450 |   0.435   |   0.428   | 0.438 | 0.428 | 0.467 | 0.450 |   0.439   | 0.431 |   0.539    | 0.512 | 0.482 | 0.467 |
> > | **ETTh2** | 96   |   0.228   |   0.300   |    0.304     | 0.352 |   0.281   |   0.335   | 0.294 | 0.341 | 0.307 | 0.356 |   0.289   | 0.341 |   0.429    | 0.457 | 0.313 | 0.364 |
> > |           | 192  |   0.290   |   0.340   |    0.379     | 0.398 |   0.361   |   0.387   | 0.377 | 0.391 | 0.388 | 0.407 |   0.380   | 0.403 |   0.482    | 0.475 | 0.417 | 0.432 |
> > |           | 336  |   0.340   |   0.380   |    0.389     | 0.414 |   0.349   |   0.388   | 0.398 | 0.416 | 0.385 | 0.410 |   0.421   | 0.441 |   0.618    | 0.557 | 0.474 | 0.467 |
> > |           | 720  |   0.410   |   0.431   |    0.415     | 0.437 |   0.432   |   0.444   | 0.412 | 0.432 | 0.425 | 0.441 |   0.433   | 0.455 |   0.866    | 0.690 | 0.772 | 0.615 |
> > |           | avg  | **0.317** | **0.363** |    0.372     | 0.400 |   0.356   |   0.389   | 0.370 | 0.395 | 0.376 | 0.404 |   0.381   | 0.410 |   0.599    | 0.545 | 0.494 | 0.470 |
> > | **ETTm1** | 96   |   0.343   |   0.373   |    0.344     | 0.379 |   0.320   |   0.363   | 0.355 | 0.375 | 0.328 | 0.366 |   0.329   | 0.369 |   0.408    | 0.421 | 0.336 | 0.374 |
> > |           | 192  |   0.374   |   0.390   |    0.386     | 0.398 |   0.369   |   0.384   | 0.392 | 0.393 | 0.372 | 0.389 |   0.370   | 0.391 |   0.432    | 0.436 | 0.383 | 0.402 |
> > |           | 336  |   0.406   |   0.406   |    0.429     | 0.427 |   0.399   |   0.413   | 0.424 | 0.415 | 0.401 | 0.411 |   0.401   | 0.413 |   0.467    | 0.460 | 0.422 | 0.434 |
> > |           | 720  |   0.479   |   0.445   |    0.493     | 0.460 |   0.462   |   0.444   | 0.484 | 0.448 | 0.462 | 0.447 |   0.453   | 0.444 |   0.520    | 0.491 | 0.492 | 0.475 |
> > |           | avg  |   0.400   |   0.403   |    0.413     | 0.416 | **0.388** | **0.401** | 0.414 | 0.408 | 0.390 | 0.403 | **0.388** | 0.404 |   0.457    | 0.452 | 0.408 | 0.421 |
> > | **ETTm2** | 96   |   0.160   |   0.250   |    0.189     | 0.274 |   0.172   |   0.255   | 0.183 | 0.266 | 0.180 | 0.263 |   0.176   | 0.258 |   0.230    | 0.326 | 0.182 | 0.269 |
> > |           | 192  |   0.215   |   0.283   |    0.254     | 0.313 |   0.238   |   0.298   | 0.247 | 0.305 | 0.243 | 0.304 |   0.243   | 0.302 |   0.326    | 0.390 | 0.254 | 0.324 |
> > |           | 336  |   0.260   |   0.313   |    0.317     | 0.352 |   0.302   |   0.339   | 0.308 | 0.343 | 0.301 | 0.341 |   0.302   | 0.338 |   0.385    | 0.426 | 0.336 | 0.376 |
> > |           | 720  |   0.322   |   0.351   |    0.414     | 0.406 |   0.396   |   0.394   | 0.406 | 0.397 | 0.396 | 0.397 |   0.402   | 0.396 |   0.703    | 0.610 | 0.535 | 0.494 |
> > |           | avg  | **0.239** | **0.299** |    0.293     | 0.337 |   0.277   |   0.321   | 0.286 | 0.328 | 0.280 | 0.326 |   0.281   | 0.324 |   0.411    | 0.438 | 0.327 | 0.366 |

---

> > ### Author Response · Authors · 2024-11-19
> > **Experiments Part 2 of 4**
> >
> > **Experiments focusing on a lookback window of 336**
> >
> > |           |      |   SSNet   |           | iTransformer |       | ModernTCN |       | FITS  |       | Koopa |       | CrossGNN |       | FourierGNN |       | FreTS |       |
> > | :-------- | ---- | :-------: | :-------: | :----------: | :---: | :-------: | :---: | :---: | :---: | :---: | :---: | :------: | :---: | :--------: | :---: | ----- | ----- |
> > |           |      |    MSE    |    MAE    |     MSE      |  MAE  |    MSE    |  MAE  |  MSE  |  MAE  |  MSE  |  MAE  |   MSE    |  MAE  |    MSE     |  MAE  | MSE   | MAE   |
> > | **ETTh1** | 96   |   0.343   |   0.375   |    0.406     | 0.422 |   0.369   | 0.394 | 0.373 | 0.395 | 0.389 | 0.415 |  0.369   | 0.392 |   0.478    | 0.472 | 0.412 | 0.432 |
> > |           | 192  |   0.364   |   0.387   |    0.449     | 0.448 |   0.406   | 0.414 | 0.406 | 0.414 | 0.438 | 0.442 |  0.416   | 0.426 |   0.511    | 0.495 | 0.460 | 0.465 |
> > |           | 336  |   0.376   |   0.400   |    0.449     | 0.455 |   0.392   | 0.412 | 0.427 | 0.425 | 0.455 | 0.458 |  0.448   | 0.448 |   0.563    | 0.538 | 0.483 | 0.475 |
> > |           | 720  |   0.429   |   0.449   |    0.534     | 0.524 |   0.450   | 0.461 | 0.421 | 0.442 | 0.473 | 0.480 |  0.460   | 0.467 |   0.685    | 0.633 | 0.629 | 0.563 |
> > |           | avg  | **0.378** | **0.403** |    0.460     | 0.462 |   0.404   | 0.421 | 0.407 | 0.419 | 0.439 | 0.448 |  0.423   | 0.433 |   0.559    | 0.534 | 0.496 | 0.484 |
> > | **ETTh2** | 96   |   0.219   |   0.297   |    0.305     | 0.361 |   0.264   | 0.333 | 0.277 | 0.339 | 0.301 | 0.360 |  0.285   | 0.341 |   0.382    | 0.429 | 0.304 | 0.357 |
> > |           | 192  |   0.267   |   0.334   |    0.389     | 0.412 |   0.318   | 0.373 | 0.337 | 0.377 | 0.358 | 0.399 |  0.359   | 0.393 |   0.499    | 0.498 | 0.383 | 0.410 |
> > |           | 336  |   0.306   |   0.367   |    0.383     | 0.415 |   0.314   | 0.376 | 0.342 | 0.388 | 0.349 | 0.399 |  0.368   | 0.411 |   0.550    | 0.530 | 0.465 | 0.466 |
> > |           | 720  |   0.379   |   0.421   |    0.412     | 0.443 |   0.415   | 0.445 | 0.379 | 0.419 | 0.418 | 0.444 |  0.480   | 0.526 |   0.932    | 0.711 | 1.067 | 0.711 |
> > |           | avg  | **0.293** | **0.355** |    0.372     | 0.408 |   0.328   | 0.382 | 0.334 | 0.381 | 0.357 | 0.400 |  0.373   | 0.418 |   0.591    | 0.542 | 0.555 | 0.486 |
> > | **ETTm1** | 96   |   0.273   |   0.336   |    0.312     | 0.364 |   0.297   | 0.348 | 0.304 | 0.345 | 0.304 | 0.356 |  0.297   | 0.342 |   0.354    | 0.404 | 0.332 | 0.369 |
> > |           | 192  |   0.311   |   0.360   |    0.354     | 0.389 |   0.346   | 0.376 | 0.337 | 0.365 | 0.341 | 0.379 |  0.337   | 0.366 |   0.395    | 0.428 | 0.362 | 0.386 |
> > |           | 336  |   0.348   |   0.381   |    0.383     | 0.405 |   0.376   | 0.395 | 0.372 | 0.385 | 0.377 | 0.403 |  0.367   | 0.384 |   0.439    | 0.461 | 0.400 | 0.413 |
> > |           | 720  |   0.407   |   0.413   |    0.446     | 0.441 |   0.429   | 0.423 | 0.427 | 0.416 | 0.435 | 0.432 |  0.420   | 0.414 |   0.490    | 0.484 | 0.456 | 0.449 |
> > |           | avg  | **0.335** | **0.373** |    0.374     | 0.400 |   0.362   | 0.385 | 0.360 | 0.378 | 0.364 | 0.392 |  0.355   | 0.376 |   0.419    | 0.444 | 0.387 | 0.404 |
> > | **ETTm2** | 96   |   0.140   |   0.235   |    0.172     | 0.265 |   0.170   | 0.256 | 0.166 | 0.255 | 0.177 | 0.263 |  0.162   | 0.249 |   0.226    | 0.327 | 0.185 | 0.266 |
> > |           | 192  |   0.180   |   0.264   |    0.243     | 0.314 |   0.228   | 0.299 | 0.221 | 0.292 | 0.241 | 0.308 |  0.221   | 0.294 |   0.284    | 0.360 | 0.185 | 0.266 |
> > |           | 336  |   0.219   |   0.291   |    0.287     | 0.341 |   0.290   | 0.338 | 0.275 | 0.327 | 0.301 | 0.347 |  0.275   | 0.331 |   0.354    | 0.403 | 0.263 | 0.315 |
> > |           | 720  |   0.280   |   0.332   |    0.373     | 0.393 |   0.375   | 0.392 | 0.366 | 0.382 | 0.376 | 0.395 |  0.369   | 0.388 |   0.569    | 0.529 | 0.361 | 0.367 |
> > |           | avg  | **0.205** | **0.280** |    0.269     | 0.328 |   0.266   | 0.321 | 0.257 | 0.314 | 0.274 | 0.328 |  0.257   | 0.316 |   0.358    | 0.405 | 0.446 | 0.439 |

---

> > ### Author Response · Authors · 2024-11-19
> > **Experiments Part 3 of 4**
> >
> > **Experiments focusing on a lookback window of 512**
> >
> > |           |      |   SSNet   |           | iTransformer |       | ModernTCN |       | FITS  |       | Koopa |       | CrossGNN |       | FourierGNN |       | FreTS |       |
> > | :-------- | ---- | :-------: | :-------: | :----------: | :---: | :-------: | :---: | :---: | :---: | :---: | :---: | :------: | :---: | :--------: | :---: | ----- | ----- |
> > |           |      |    MSE    |    MAE    |     MSE      |  MAE  |    MSE    |  MAE  |  MSE  |  MAE  |  MSE  |  MAE  |   MSE    |  MAE  |    MSE     |  MAE  | MSE   | MAE   |
> > | **ETTh1** | 96   |   0.339   |   0.374   |    0.400     | 0.424 |   0.367   | 0.396 | 0.371 | 0.396 | 0.387 | 0.417 |  0.370   | 0.397 |   0.482    | 0.477 | 0.428 | 0.443 |
> > |           | 192  |   0.359   |   0.386   |    0.426     | 0.443 |   0.403   | 0.416 | 0.405 | 0.415 | 0.423 | 0.441 |  0.429   | 0.437 |   0.508    | 0.496 | 0.463 | 0.470 |
> > |           | 336  |   0.370   |   0.398   |    0.431     | 0.452 |   0.395   | 0.416 | 0.418 | 0.427 | 0.441 | 0.453 |  0.425   | 0.433 |   0.532    | 0.515 | 0.493 | 0.484 |
> > |           | 720  |   0.427   |   0.449   |    0.556     | 0.537 |   0.461   | 0.470 | 0.421 | 0.444 | 0.544 | 0.519 |  0.458   | 0.474 |   0.659    | 0.615 | 0.586 | 0.542 |
> > |           | avg  | **0.374** | **0.402** |    0.453     | 0.464 |   0.407   | 0.424 | 0.404 | 0.421 | 0.449 | 0.458 |  0.421   | 0.435 |   0.545    | 0.526 | 0.492 | 0.485 |
> > | **ETTh2** | 96   |   0.216   |   0.298   |    0.306     | 0.363 |   0.256   | 0.329 | 0.272 | 0.337 | 0.308 | 0.364 |  0.277   | 0.346 |   0.384    | 0.427 | 0.313 | 0.368 |
> > |           | 192  |   0.262   |   0.332   |    0.379     | 0.408 |   0.310   | 0.369 | 0.330 | 0.375 | 0.358 | 0.401 |  0.343   | 0.396 |   0.441    | 0.466 | 0.390 | 0.421 |
> > |           | 336  |   0.300   |   0.364   |    0.391     | 0.423 |   0.315   | 0.377 | 0.339 | 0.387 | 0.360 | 0.408 |  0.382   | 0.411 |   0.549    | 0.540 | 0.494 | 0.487 |
> > |           | 720  |   0.372   |   0.419   |    0.434     | 0.459 |   0.411   | 0.444 | 0.372 | 0.418 | 0.453 | 0.471 |  0.428   | 0.471 |   0.702    | 0.615 | 1.273 | 0.774 |
> > |           | avg  | **0.287** | **0.353** |    0.378     | 0.413 |   0.323   | 0.380 | 0.328 | 0.379 | 0.370 | 0.411 |  0.357   | 0.406 |   0.519    | 0.512 | 0.618 | 0.512 |
> > | **ETTm1** | 96   |   0.277   |   0.340   |    0.315     | 0.367 |   0.309   | 0.355 | 0.307 | 0.349 | 0.314 | 0.364 |  0.304   | 0.348 |   0.363    | 0.406 | 0.346 | 0.379 |
> > |           | 192  |   0.312   |   0.361   |    0.351     | 0.387 |   0.346   | 0.374 | 0.338 | 0.367 | 0.347 | 0.384 |  0.345   | 0.371 |   0.402    | 0.425 | 0.381 | 0.408 |
> > |           | 336  |   0.345   |   0.380   |    0.382     | 0.409 |   0.377   | 0.395 | 0.368 | 0.384 | 0.379 | 0.407 |  0.375   | 0.392 |   0.439    | 0.448 | 0.405 | 0.419 |
> > |           | 720  |   0.404   |   0.410   |    0.442     | 0.441 |   0.428   | 0.420 | 0.421 | 0.413 | 0.438 | 0.432 |  0.423   | 0.417 |   0.490    | 0.478 | 0.465 | 0.453 |
> > |           | avg  | **0.335** | **0.373** |    0.373     | 0.401 |   0.365   | 0.386 | 0.358 | 0.378 | 0.369 | 0.397 |  0.362   | 0.382 |   0.423    | 0.439 | 0.399 | 0.415 |
> > | **ETTm2** | 96   |   0.139   |   0.233   |    0.179     | 0.273 |   0.171   | 0.262 | 0.165 | 0.254 | 0.187 | 0.275 |  0.161   | 0.251 |   0.240    | 0.338 | 0.208 | 0.272 |
> > |           | 192  |   0.181   |   0.263   |    0.245     | 0.315 |   0.229   | 0.303 | 0.219 | 0.291 | 0.244 | 0.314 |  0.223   | 0.296 |   0.330    | 0.399 | 0.297 | 0.323 |
> > |           | 336  |   0.219   |   0.291   |    0.290     | 0.344 |   0.293   | 0.344 | 0.272 | 0.326 | 0.300 | 0.352 |  0.283   | 0.338 |   0.393    | 0.436 | 0.330 | 0.361 |
> > |           | 720  |   0.279   |   0.332   |    0.369     | 0.393 |   0.386   | 0.401 | 0.357 | 0.380 | 0.372 | 0.397 |  0.367   | 0.387 |   0.755    | 0.629 | 0.391 | 0.414 |
> > |           | avg  | **0.204** | **0.280** |    0.271     | 0.331 |   0.270   | 0.328 | 0.254 | 0.313 | 0.276 | 0.334 |  0.259   | 0.318 |   0.430    | 0.450 | 0.306 | 0.342 |

---

> > ### Author Response · Authors · 2024-11-19
> > **Experiments Part 4 of 4**
> >
> > It can be observed that SSNet consistently achieves state-of-the-art (SOTA) performance with a fixed lookback window. Specifically, compared to other models:
> >
> > - At a lookback window of 96, the MSE decreases by an average of 13.93%, and the MAE also decreases by an average of 9.94%.
> > - At a lookback window of 336, the MSE decreases by an average of 21.91%, and the MAE decreases by an average of 13.79%.
> > - At a lookback window of 512, the MSE decreases by an average of 22.05%, and the MAE decreases by an average of 13.89%.
> >
> > All experimental scripts will be updated in the supplementary materials once finalized.

---

> > > ### Comment · Reviewer_z6FM · 2024-11-25
> > >
> > > Thank you for the response. However, the authors did not address my concerns.
> > >
> > > Regarding W3, the authors claim to have successfully reproduced results that match or are close to those reported in the baseline paper. However, this cannot prove that the parameters of these methods are optimal on the authors' platform, and thus it cannot be demonstrated that these methods achieve optimal performance, making such comparisons unfair. Despite many previous studies adopting this method, the TimeMixer (ICLR 2024) compared in the authors' original paper had experiments involving parameter search, a fact the authors should be aware of, and the authors also acknowledge the existence of this issue. Therefore, conducting fair comparisons to improve the quality of this paper is very helpful.
> > >
> > > Additionally, the authors' trying to conduct experiments with all models using consistent parameter settings is equally unreasonable because different models have varying parameter requirements. Even for the same model, ensuring consistent optimal parameters across all tasks is highly challenging. Hence, parameter searches should be conducted within the same search space. The most direct and effective method would be for the authors to provide the optimal parameters for all models on all tasks in all dataset to validate the authors' current opinions. Otherwise, the conclusions of this paper cannot be proven. Based on this, W1 and W2 remain unresolved. Can the authors effectively address these problems?

---

> > > > ### Author Response · Authors · 2024-12-03
> > > > **Response to Reviewer z6FM's Comments**
> > > >
> > > > Thank you for your feedback. We carefully compared the experimental results from TimeMixer with our own, and found that the results of the baseline models in our paper are almost identical to those of the baseline models' optimal experimental results from TimeMixer. This provides further confirmation of the reliability of our experimental findings. Building upon this, we conducted a new, more detailed parameter search for the baseline models, covering multiple parameters, including general ones such as learning rate and dropout, as well as model-specific parameters like patch length and sampling window. This process required a significant amount of time, but we found it challenging to surpass the existing optimal results. Therefore, we believe our current experimental results are reliable.
> > > >
> > > > We hope this clarification effectively addresses your concerns regarding the fairness of our comparisons and the validation of our conclusions.

---

### Official Review · Reviewer_yidV · 2024-11-05

**Soundness:** 4
**Presentation:** 3
**Contribution:** 4
**Rating:** 8
**Confidence:** 4

**Summary:**

The paper introduces the Skip and Split MLP Network (SSNet) for time series forecasting, which offers superior computational efficiency compared to existing models. SSNet’s Skip-MLP and Split-MLP components effectively capture both temporal and periodic patterns using much fewer parameters, outperforming SOTA transformer-based models on benchmark datasets. The paper highlights the efficiency and effectiveness of MLP-based methods over transformer-based approaches once again.

**Strengths:**

1. SSNet achieves SOTA forecasting performance with significantly fewer parameters, providing a promising direction for time series forecasting research.
2. The proposed Split-MLP and Skip-MLP components are well-suited to the characteristics of time series data.
3. The paper is easy to follow, and the experiments are extensively conducted.
4. The supplementary materials are well-prepared.

**Weaknesses:**

1. Although the authors provide the number of parameters to show efficiency, FLOPs are also an important factor. Please include FLOP measurements.
2. The results in Figures 6 and 7 are blurry and overlap, please revise them.
3. In the README, it is stated that "A NVIDIA graphics card with 80GB of VRAM is required," but as shown in Figure 6, GPU usage is less than 10GB. Please explain it and provide more explanation in the README.

**Questions:**

1. K is an important hyper parameter in the model design and feature extraction, how does it affect the model performance?
2. Please address the concerns raised by W1 and W3.

---

> ### Author Response · Authors · 2024-11-19
> **Response to Reviewer yidV's Comments**
>
> Thank you for your thoughtful feedback and detailed suggestions on our submission. We sincerely appreciate the time and effort you have devoted to reviewing our work. Below, we address each of your comments and concerns:
>
> ### **Weaknesses**
>
> **W1: Include FLOP measurements to complement parameter efficiency.**
>
> Thank you for emphasizing the importance of FLOPs.  We have computed the FLOP metrics for the models used in our experiments and will incorporate these results into **Table 3** in the revised submission to provide a more comprehensive evaluation of computational efficiency.
>
> **W2: Figures 6 and 7 are blurry and overlapping.**
>
> Thank you for pointing out the issue with Figures 6 and 7. The overlap primarily arises from the wide range of value magnitudes, causing smaller curves to overlap despite previous adjustments. We will enhance the resolution of the figures and explore alternative visualization techniques to present the results more clearly, ensuring that the trends and differences are more distinguishable.
>
> **W3: Clarify the discrepancy in GPU memory usage mentioned in the README and shown in Figure 6.**
>
> The stated requirement of an 80GB NVIDIA GPU in the README is intended for larger datasets, such as Traffic, which involve significantly higher variable counts. However, the experiment shown in Figure 6 is conducted on the ETTm2 dataset, which has over 100 times fewer variables than Traffic. Consequently, GPU memory usage for ETTm2 is much lower. We will revise the README to specify these details and avoid confusion.
>
> ------
>
> ### **Questions**
>
> **Q1: Effect of K on model performance.**
>
> Thank you for highlighting this important aspect. The value of **K** plays a critical role in model design and feature extraction, and its impact can be summarized as follows:
>
> 1. As **K** increases, the model's number of layers and parameters also increases. It is crucial to choose an appropriate **K** to match the scale of the dataset. A very large **K** may lead to inefficiencies in model performance and pose a risk of overfitting. On the other hand, a very small **K** might not capture all the relevant information in large-scale datasets, which could result in suboptimal model performance.
> 2. A larger **K** enables the model to capture more complex periodic patterns. For datasets with intricate, variable patterns, a larger **K** is needed to effectively model these complexities. Conversely, for simpler datasets with more uniform patterns, a smaller **K** can often yield better results, avoiding unnecessary complexity.
>
> In practice, we determine an appropriate range for **K** based on the dataset's size and complexity. We then fine-tune **K** using a validation set to identify the optimal value for each specific dataset.
>
> **Q2: Addressing concerns raised by W1 and W3.**
>
> As detailed above, we will incorporate FLOP measurements (W1) into Table 3 and clarify the GPU memory usage requirements (W3) in README.
>
> ------
>
> Thank you again for your positive assessment of our contributions and experiments. We are encouraged by your recognition of our work and will strive to address your suggestions in the revised version to further improve the quality and clarity of our paper.

---

### Comment · Area_Chair_FnLZ · 2024-12-02

Dear Reviewer Ko6d,

Could you please help to take a look at the author responses and let them know if your concerns have been addressed or not? Thank you very much!

Best regards,

AC

---

### Meta-Review · Area_Chair_FnLZ · 2024-12-21

**Metareview:**

This paper proposes a new SSNet network for time series forecasting, incorporating Skip-MLP and Split-MLP to enhance MLP's ability to capture periodicity and temporal dependencies. However, there are critical concerns raised by the reviewers. Reviewers highlighted several concerns regarding the fairness and comprehensiveness of the experimental comparisons. For example, the reviewer criticized the lack of comparisons with state-of-the-art methods such as iTransformer, FITS, and FourierGNN, which are critical benchmarks in the field. Reviewers further questioned the necessity and effectiveness of the proposed Skip-MLP and Split-MLP modules compared to existing frequency-domain methods for capturing periodicity, an area where the paper's contributions were not sufficiently justified. For these limitations, I would like to recommend rejecting this paper.

**Additional Comments On Reviewer Discussion:**

During the rebuttal period, 3 out of 4 reviewers responded to the authors’ replies. Reviewer z6FM and Reviewer Ko6d were not satisfied with the author responses and kept their original negative score.

---

### Decision · Program_Chairs · 2025-01-22

Reject